# Assessing the Water Status and Leaf Pigment Content of Olive Trees: Evaluating the Potential and Feasibility of Unmanned Aerial Vehicle Multispectral and Thermal Data for Estimation Purposes

Pedro Marques [1,2] , Luís Pádua [2,3,4] , Joaquim J. Sousa [4,5] and Anabela Fernandes-Silva [1,2,3,*]

1   Agronomy Department, School of Agrarian and Veterinary Sciences, University of Trás-os-Montes e Alto Douro, 5000-801 Vila Real, Portugal; pedro.marques@utad.pt
2   Centre for the Research and Technology of Agro-Environmental and Biological Sciences, University of Trás-os-Montes e Alto Douro, 5000-801 Vila Real, Portugal; luispadua@utad.pt
3   Institute for Innovation, Capacity Building and Sustainability of Agri-Food Production, University of Trás-os-Montes e Alto Douro, 5000-801 Vila Real, Portugal
4   Engineering Department, School of Science and Technology, University of Trás-os-Montes e Alto Douro, 5000-801 Vila Real, Portugal; jjsousa@utad.pt
5   Centre for Robotics in Industry and Intelligent Systems (CRIIS), INESC Technology and Science (INESC-TEC), 4200-465 Porto, Portugal
*   Correspondence: anaaf@utad.pt

**Abstract:** Global warming presents a significant threat to the sustainability of agricultural systems, demanding increased irrigation to mitigate the impacts of prolonged dry seasons. Efficient water management strategies, including deficit irrigation, have thus become essential, requiring continuous crop monitoring. However, conventional monitoring methods are laborious and time-consuming. This study investigates the potential of aerial imagery captured by unmanned aerial vehicles (UAVs) to predict critical water stress indicators—relative water content (RWC), midday leaf water potential ($\Psi_{MD}$), stomatal conductance ($g_s$)—as well as the pigment content (chlorophyll ab, chlorophyll a, chlorophyll b and carotenoids) of trees in an olive orchard. Both thermal and spectral vegetation indices are calculated and correlated using linear and exponential regression models. The results reveal that the thermal vegetation indices contrast in estimating the water stress indicators, with the Crop Water Stress Index (CWSI) demonstrating higher precision in predicting the RWC ($R^2 = 0.80$), $\Psi_{MD}$ ($R^2 = 0.61$) and $g_s$ ($R^2 = 0.72$). Additionally, the Triangular Vegetation Index (TVI) shows superior accuracy in predicting the chlorophyll ab ($R^2 = 0.64$) and chlorophyll a ($R^2 = 0.61$), while the Modified Chlorophyll Absorption in Reflectance Index (MCARI) proves most effective for estimating the chlorophyll b ($R^2 = 0.52$). This study emphasizes the potential of UAV-based multispectral and thermal infrared imagery in precision agriculture, enabling assessments of the water status and pigment content. Moreover, these results highlight the vital importance of this technology in optimising resource allocation and enhancing olive production, critical steps towards sustainable agriculture in the face of global warming.

**Keywords:** irrigation management; precision agriculture; vegetation indices; chlorophyll content; carotenoid content; water stress indicators

## 1. Introduction

Water scarcity poses a significant challenge to socio-economic development and ranks as the most critical current risk according to the World Economic Forum's annual risk report [1]. Notably, agricultural irrigation is responsible for 70% of global freshwater consumption, increasing the competition between agriculture and other economic sectors [2]. Consequently, adopting water-efficient agricultural systems and irrigation strategies becomes imperative for ensuring sustainable production.

Precision irrigation is a strategic approach aimed at enhancing the agricultural yield while simultaneously conserving valuable water resources and mitigating adverse environmental impacts. This topic is of particular significance, especially for developing nations and small-scale agricultural farms [3]. However, the advanced nature of precision irrigation technology from developed countries presents substantial challenges in countries such as China, Brazil and India, including technology disparities, real-time data processing limitations and high implementation costs. Given that small-scale farms contribute significantly to irrigated agriculture in developing nations and are expected to play an even larger role in the future, it is imperative to develop cost-effective, data-driven approaches for water management to support socio-economic development [4]. The adoption of balanced precision irrigation techniques is expected, as addressed to the specific socio-economic conditions prevailing in developing countries and small-scale farming contexts. These practices offer opportunities to enhance agricultural irrigation efficiency in such regions while mitigating adverse environmental impacts.

In the specific case of olive trees, numerous irrigation strategies have been developed and assessed to improve water use efficiency, with a primary focus on enhancing yield losses and potentially improving olive oil quality [5]. Based on the physiological response of crops to water stress, the adoption of deficit irrigation (DI) has been the subject of substantial focus in various investigations [6–9]. DI, a rational water-saving approach, subjects crops to controlled water stress levels in order to optimise water productivity while minimising the impact on the yield [10]. Primarily, two DI strategies have been extensively studied in olive trees: regulated deficit irrigation (RDI) and sustained deficit irrigation (SDI) [10,11]. The divergence between these strategies lies in the fact that RDI ensures the absence of water stress during phenological phases sensitive to water scarcity, imposing water stress exclusively during the pit-hardening phase through the reduction or cessation of irrigation. Conversely, SDI imposes a consistent water restriction across the entire irrigation season, maintaining a uniform stress level throughout. In the context of olive growing in Portugal, both RDI and SDI have been studied, providing valuable insights that demonstrate their ability to increase the net farm income, enhance the yield, reduce costs and conserve water [6,10,12]. However, to successfully implement these methods in practice and minimise any negative impact on the yield, it is crucial to thoroughly understand how the crops react to water stress. This knowledge, combined with continuous monitoring of water stress, becomes exceptionally important. Commonly adopted water status indicators include the relative water content (RWC), leaf water potential at predawn ($\Psi_{PD}$) or midday ($\Psi_{MD}$), and stomatal conductance ($g_s$).

Conversely, indirect methods for monitoring water stress in plants can also be performed, including the quantification of the photosynthetic pigment content of leaves, such as the total chlorophyll (Chl *ab*), chlorophyll a (Chl *a*), chlorophyll b (Chl *b*), and carotenoids. The content of these pigments within leaves has a strong correlation with the plant's physiological condition, photosynthetic efficiency, developmental stage, productivity and level of water stress [13]. Higher pigment concentrations are typically evident in non-stressed plants characterised by optimal growth, in contrast to those experiencing stress due to abiotic and biotic factors [14].

The conventional methods for determining the water status indicators and leaf pigment content frequently require labour-intensive and invasive procedures. Quantifying the water status indicators demands specialised equipment, such as Scholander pressure chambers and porometers, while analysing the pigment content demands costly methodologies such as spectrophotometry and high-performance liquid chromatography [15]. Nevertheless, these methods possess a significant limitation as they focus solely on individual samples, offering restricted insights from single leaves or branches without considering the entirety of the crop. As a result, an urgent requirement arises for more efficient tools to support decision-support systems for farmers and agronomists.

Recently, emerging technologies such as remote sensing have shown significant potential in optimising agricultural practices and resource allocation to enhance production

efficiency while reducing costs and mitigating yield losses [16]. The use of remote sensing for crop monitoring through satellites or unmanned aerial vehicles (UAVs) has gained prominence across extensive geographical domains [17], including several crops, both annual and perennial. Examples include olive trees [18], chestnut trees [19], lemon trees [20], vineyards [21], corn [22], winter wheat [23], lettuce [24], Bermuda grass [25], barley [26], and others. This technology provides data that can be processed to formulate vegetation indices (VIs) for various applications, including inventory management, dendrometric estimations, identification and monitoring of plant biotic stress, irrigation optimisation and soil properties estimation [16]. The efficacy of this methodology lies in its capacity to infer fluctuations in crop vigour, density, water status, and productivity.

Over the last few decades, this technological innovation has gained considerable adherence, particularly in countries where economic activities are interlinked with crop cultivation. An example of this phenomenon is observed in Mediterranean countries, where the olive tree stands as a prominent crop. The olive tree (*Olea europaea* L.) is a crucial crop within the Mediterranean region, playing a substantial role in the economic development of these nations [27]. In Portugal, olive cultivation extents over an expansive 373,500 hectares and includes about 120,000 farms [28]. The country ranks seventh globally and fourth within the European Union in olive oil production, generating over 100,000 tonnes of this product [29]. Notably, olive oil constitutes about 7% of Portugal's agricultural exports, which have demonstrated remarkable growth since 2000.

Traditionally, olive trees were cultivated under rainfed conditions, given their robust resilience to drought and ability to weather water stress [30]. Despite this notable drought tolerance, the growth, development and yield of olive trees remained primarily influenced by atmospheric circumstances [31], particularly precipitation and air temperature. Within this traditional olive production system, an alternating cycle of high and low yields emerged, a pattern that has been intensified by global warming, leading to extended periods of hot and dry seasons. These conditions have adversely affected the productivity of olive trees.

To enhance productivity and establish consistent yield patterns, the imperative of irrigating olive orchards has gained dominant importance. This significance has been underscored by numerous studies conducted across diverse regions [32]. In several areas, the allocation of land for irrigating olive orchards has been expanding to respond to the adverse complications of drought and to increase olive oil production, aligning with the rising demand for this product due to its recognised health benefits [33]. Consequently, remote sensing techniques have found extensive application in optimising irrigation management for this specific agricultural cultivation.

In the literature, UAVs equipped with thermal and hyperspectral sensors are commonly used for monitoring the plant water status and estimating the chlorophyll content. Egea et al. [34] established correlations between the thermal Crop Water Stress Index (CWSI) and physiological parameters, including the $\Psi_{MD}$, $g_s$ and leaf transpiration rate, in super-high-density olive orchards. The CWSI showed notable suitability in monitoring water stress and assessing the water status variability, with a coefficient of determination ($R^2$) over 0.6. In a different investigation, Ben-Gal et al. [35] explored the disparity between empirical and analytical CWSI methodologies for estimating the $\Psi_{MD}$ and $g_s$. Remarkably, both approaches yielded congruent outcomes. However, the analytical CWSI showcased a distinct advantage due to its relative practicality, with higher correlation between the canopy temperature and CWSI ($R^2 > 0.8$), but with a weak correlation with water stress indicators such as the $\Psi_{MD}$ and stomatal resistance ($R^2 < 0.5$). More recently, Caruso et al. [36] conducted an analysis employing thermal images captured by a UAV to monitor the water status, canopy growth and yield of "Frantoio" and "Leccino" olive cultivars under different irrigation regimes. The CWSI derived from thermal imaging showed a consistent relationship with the $\Psi_{MD}$ ($R^2 = 0.83$) across both cultivars. Effectively characterising the seasonal water status and discerning differences between irrigation strategies, the CWSI emerged as a powerful tool. Moreover, significant associations were identified between

the CWSI and the canopy growth, fruit yield and oil yield, underscoring its utility as an indicator of olive trees' water status. These findings emphasise the crucial role of the CWSI in evaluating irrigation management and its consequent impact on crop productivity. In a distinctive approach using different wavelengths, Marino et al. [37] investigated the potential of the Photochemical Reflectance Index (PRI) [38], Water Index (WI) [39], and Normalised Difference Vegetation Index (NDVI) [40] to detect plant responses to seasonal drought. The authors concluded that these indices effectively discriminate the effects of drought on rainfed trees and their subsequent recovery. Notably, the WI exhibited the most robust correlation with the photosynthesis rate ($R^2 = 0.62$), $g_s$ ($R^2 = 0.59$), and sap flux density ($R^2 = 0.69$), while the NDVI displayed stronger relationships with the $\Psi_{MD}$ ($R^2 = 0.68$). While hyperspectral sensors involve higher costs compared to multispectral (MSP) sensors, Zarco-Tejada et al. [41] effectively employed them to estimate the Chl *ab* content. The most favourable performance was achieved through the ratio of the Modified Chlorophyll Absorption Ratio Index (MCARI) [42] to the Optimised Soil Adjusted Vegetation Index (OSAVI) [43], resulting in an impressive $R^2$ value of 0.69.

However, the aforementioned studies have exclusively applied singular methodologies to compute the CWSI and have assessed only a subset of the core optical indices. Furthermore, none of these investigations have established correlations among the diverse indices and the RWC—an indispensable and precise water status indicator. Additionally, the existing literature predominantly focuses on the analysis of the Chl *ab* content. To overcome the identified limitations of prior investigations, the principal aim of this study is to assess the viability of several spectral and thermal VIs derived from UAV-acquired imagery to estimate water status indicators (RWC, $\Psi_{MD}$, and $g_s$) and physiological performance through an analysis of the leaf pigment content (Chl *ab*, Chl *a*, Chl *b*, and carotenoids) within olive trees cultivated under field conditions in the northeast region of Portugal. Additionally, our investigation includes a comprehensive examination of two thermal VIs (CWSI and Stomatal Conductance Index—$I_g$) and a list of 37 optical VIs. Additionally, five distinct methodologies are employed to compute the CWSI, while three distinct methods are applied to calculate the $I_g$. To accomplish this, we analyse a series of linear and exponential regression models to predict the aforementioned parameters.

## 2. Materials and Methods

### 2.1. Study Site, Climate and Soil Characterisation

This study was conducted during three consecutive years (2019 to 2021) in a 2 ha irrigated olive orchard (Cv. "Cobrançosa") located in the Vilariça Valley (Vilarelhos: 41.33°N, 7.04°W; 240 m altitude), a representative olive cultivation region in northeast Portugal, which belongs to the olive oil denomination "DOP—Azeite de Trás-os-Montes". The experiment involved 25-year-old olive trees spaced at 6 m × 6 m (278 trees/ha) and showing uniform characteristics: 4 m in height, 52 cm in trunk diameter and a crown diameter of 3.8 m, corresponding to an average crown area of 11 m$^2$.

The study site experiences a Mediterranean climate, classified as Csa according to the Köppen climate system [44]. Meteorological parameters, including the solar radiation, air temperature, relative humidity, rainfall and wind speed were collected by an adjacent automated weather station. The highest average annual temperature was 16.4 °C in 2020, contrasting with the lowest of 15.5 °C in 2019 (Figure 1). During the summer months, July 2019 reached the highest average monthly temperature of 25.6 °C, July 2020 reached 28.4 °C, and August 2021 recorded 25.7 °C. The peak absolute daily temperatures were 40.9 °C (DOY 232) in 2019, 40.0 °C (DOY 219) in 2020, and 41.9 °C (DOY 225) in 2021. The annual precipitation and reference evapotranspiration (ET$_0$) were, respectively, 744 mm and 1004 mm in 2019, 660 mm and 981 mm in 2020, and 519 mm and 968 mm in 2021. Throughout the irrigation period (June–October), the effective rainfall (calculated as 75% of the daily rainfall) and ET$_0$ measured 61.0 mm and 547.0 mm in 2019, 58.7 mm and 559.8 mm in 2020, and 96.5 mm and 542.3 mm in 2021.

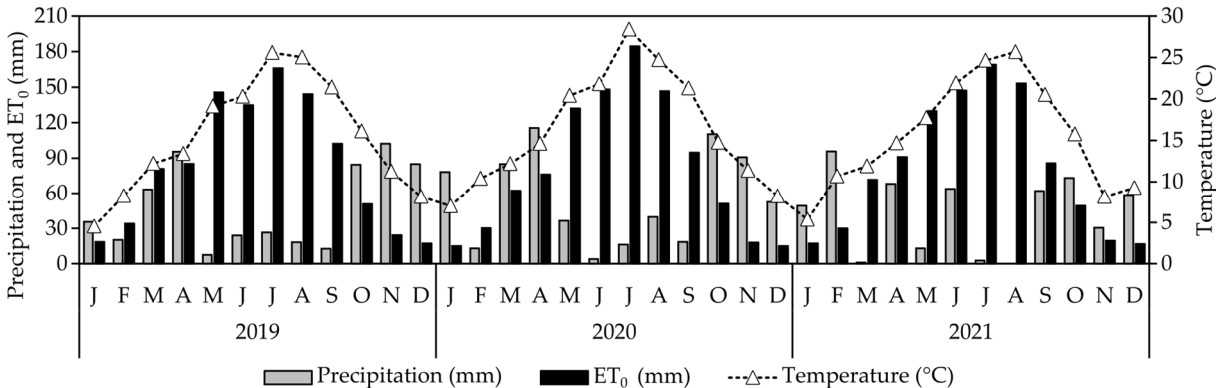

**Figure 1.** Monthly values of the reference evapotranspiration (ET$_0$, mm), rainfall (mm) and mean monthly air temperature (°C) observed at the study site from 2019 to 2021.

The soil characteristics were determined in 2019 via sampling at three depths (0–0.2; 0.2–0.4 m and 0.4–0.60 m). The soil was classified as Eutric Leptosols developed on metamorphic rocks (schists) of sandy loam [10]. The organic matter content in the soil was low (10 g kg$^{-1}$), and the pH was 6.6 and the apparent bulk density varied from 1.29 t m$^{-3}$ in the topsoil layer (0–0.20 m) to 1.26 t m$^{-3}$ in the deepest layer (0.60 m), with an overall average of 1.30 t m$^{-3}$.

### 2.2. Irrigation Treatments and Experimental Design

The experimental layout included seven adjacent blocks, each consisting of three rows. Only the central trees were selected for sampling, and they were subjected to seven distinct irrigation regimes (Figure 2) from late spring to early autumn (June–October). The water application was determined through the computation of the crop evapotranspiration (ET$_c$), as calculated using the FAO method [45], expressed by $ET_c = ET_0 \times K_c$. The estimation of the crop coefficient (K$_c$) during the irrigation period was estimated by implementing the model outlined by Orgaz et al. [46] on a weekly basis. The irrigation treatments included the following: (1) full irrigation (FI$_{100}$), defined as the control treatment, receiving 100% of the estimated evapotranspiration (ET$_c$); (2) over full irrigation (FI$_{120}$), receiving 120% of the estimated ET$_c$; (3) two sustained deficit irrigation regimes (SDI$_{60}$ and SDI$_{30}$), respectively receiving 60% and 30% water of FI$_{100}$ and sustained throughout the irrigation season; (4) two regulated deficit irrigation regimes (RDI$_{100}$ and RDI$_{60}$), with RDI$_{100}$ being irrigated equivalently to FI$_{100}$ except during pit-hardening, when the irrigation was decreased to 10%, and RDI$_{60}$ being irrigated as SDI$_{60}$ except during pit-hardening, when the irrigation was ceased; and (5) farmer irrigation (FMI) with 8 h of irrigation twice weekly. Drip irrigation was scheduled daily across all seven regimes, with the drippers spaced at 1 m intervals, each with a flow rate of 4 L h$^{-1}$. For the assessment of the water status indicators, the pigment content quantification and the estimation of the average values for several VIs, a random selection of five olive trees were chosen along the central line of each irrigation regime.

### 2.3. Soil Water Content and Plant Water Status Indicator Measurements

To monitor the volumetric soil water content ($\theta_v$), six access tubes were vertically inserted into the soil per irrigation treatment (excluding the FMI). These tubes were placed within three olive trees (two per tree) to a depth of approximately 80 cm, positioned at 30 cm from the trunk and adjacent to the drippers. The measurements were performed from an initial depth of 20 cm, progressing with 20 cm intervals. For this purpose, a TDR T3 probe (TRIME-FM, IMKO GmbH, Ettlingen, Germany) was used. However, due to logistical limitations concerning potential access tubes damage, these measurements were unattainable in 2021.

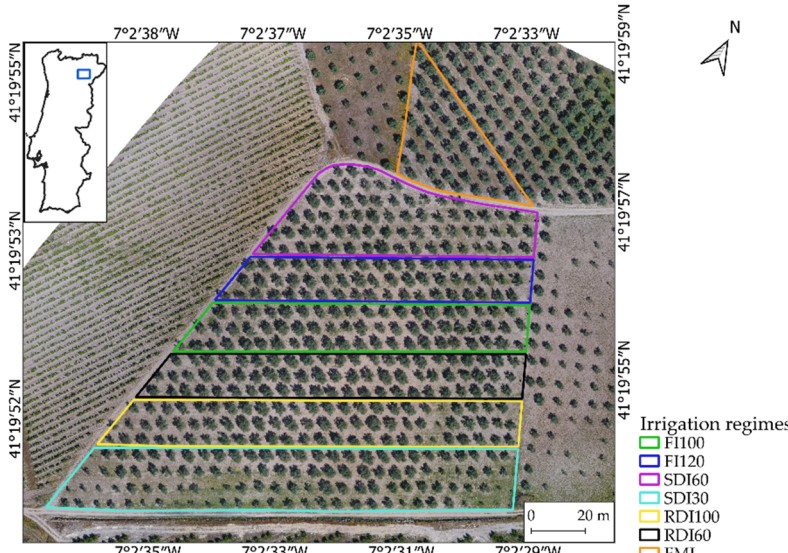

**Figure 2.** Layout of the irrigation experiment in the olive orchard (Cv. Cobrançosa) submitted to different irrigation regimes: full irrigated 100% ($FI_{100}$), over full irrigated 120% ($FI_{120}$), sustained deficit irrigation 60% ($SDI_{60}$), sustained deficit irrigation 30% ($SDI_{30}$), regulated deficit irrigation 100% ($RDI_{100}$), regulated deficit irrigation 60% ($RDI_{60}$) and farmer managed irrigation (FMI), during 2019–2021, located in the northeast of Portugal (41.33°N, 7.04°W).

The water status of the olive trees was determined through the measurement of the $\Psi_{MD}$. For this purpose, young leafy shoots were collected from sunlit positions within the crowns of three representative olive trees within each irrigation treatment. These shoots were promptly sealed within a plastic bag to prevent water loss and then transferred to a PMS 1000 pressure chamber (PMS Instrument, Albany, NY, USA). Additionally, the $g_s$ was measured at midday within the same olive trees. Using an SC-1 porometer (Decagon Devices Inc., Washington, DC, USA), measurements were taken from two leaves per each of the three olive trees. Each measurement derived from a one-year-old leaf, selected from a sunny crown position, and the porometer was set to auto mode for a 30 s interval.

Furthermore, the RWC, a method associated with reduced time and labour intensiveness compared to the $\Psi_{MD}$ and $g_s$, was assessed. Five olive trees from each irrigation treatment were used for this evaluation. From each chosen olive tree, three one-year-old leaves were detached, sealed within glass tubes, preserved in a cooled container and transported to the laboratory. Estimation of the RWC followed the protocol outlined by Marques et al. [47].

### 2.4. Quantification of Chlorophyll a, b and Total Carotenoid Pigment Content

Leaf pigment samples were obtained on DOY 199 (2019) and DOY 207 (2021). In each irrigation treatment, ten leaves of identical age were collected from the same set of five previously selected olive trees. These leaves were promptly placed within a refrigerated container, ensuring preservation during transport to the laboratory, and subsequently stored at −80 °C for biochemical analysis. The quantification of the chlorophyll and carotenoid pigments was conducted following the methodologies outlined in [48], with necessary adaptations.

To extract the pigments, 100 mg of dried leaf mass was introduced into centrifuge tubes, into which 5 mL of 80% acetone was added. The samples were shaken at 30 Hz for five minutes, followed by centrifugation for five minutes (4000 rpm). Then, 200 μL of supernatant from each sample was transferred into a 96-well microplate and the absorbance was measured at 470, 645, and 662 nm against a blank using a microplate reader.

The obtained absorbance values were adjusted to the path-length-corrected absorbance for 0.5 cm, as outlined by Marques et al. [47]. The pigment content was calculated based on this corrected pathlength. All the analytical procedures were conducted in triplicate.

### 2.5. Remote Sensing Data Collection

Remote sensing data were collected using two UAVs: a fixed-wing UAV, the eBee (senseFly SA, Lausanne, Switzerland), and a rotary-wing UAV, the Phantom 4 (DJI, Shenzhen, China). The Phantom 4 was equipped with a Parrot Sequoia sensor, enabling the acquisition of MSP data spanning the green (530–570 nm), red (640–680 nm), red-edge (730–740 nm), and near-infrared—NIR (770–810 nm) spectral ranges, with a resolution of 1.2 MP. RGB imagery was also acquired for visualisation purposes using the 12.4 MP camera on the Phantom 4. Radiometric calibration was achieved using in-flight irradiance data, with the reflectance data obtained pre-flight using a calibration target.

For the thermal infrared (TIR) imagery acquisition, the eBee UAV equipped with a thermoMAP sensor was used, collecting data within 7500 nm to 13,500 nm, with a resolution of $640 \times 512$. The UAV flights were performed from May to September between 2018 and 2021, excluding 2020 due to UAV technical issues, limiting the thermal flights to 2018, 2019, and 2021. Both thermal flights and ground measurements of the water status indicators were conducted at midday. For the MSP and RGB flights, scheduling near solar noon was chosen to minimise the shadow incidence.

The UAV flight campaigns from both UAVs were performed at an 80 m flight height, with a 90% longitudinal overlap and 70% lateral overlap. This configuration produced data with spatial resolutions of around 0.03 m, 0.16 m, and 0.07 m, respectively, for the RGB, TIR, and MSP data. Table 1 summarises the performed flight campaigns.

**Table 1.** UAV flight campaigns for the multispectral and thermal data acquisition. TIR: thermal infrared; MSP: multispectral.

| DOY | Model Applicability | Data Type | Platform |
|---|---|---|---|
| 253 (2018) | Prediction | TIR | eBee |
| 199 (2019) | Regression | RGB and MSP<br>TIR | Phantom 4<br>eBee |
| 261 (2019) | Regression | RGB and MSP | Phantom 4 |
| 145 (2020) | Regression | RGB and MSP | Phantom 4 |
| 189 (2020) | Prediction | RGB and MSP | Phantom 4 |
| 207 (2021) | Regression | RGB and MSP<br>TIR | Phantom 4<br>eBee |

### 2.6. UAV Data Processing

To ensure the precise alignment of the RGB, MSP, and TIR imagery, aluminium targets were strategically positioned across the olive orchard. The data were processed using Pix4Dmapper Pro (Pix4D SA, Lausanne, Switzerland), using structure-from-motion (SfM) algorithms to construct a high-density point cloud from the captured images. Using inverse distance weighting (IDW), the point cloud was interpolated, producing orthorectified raster outputs, including the digital surface model (DSM), digital terrain model (DTM) and reflectance maps of the MSP bands. Additionally, the canopy height model (CHM) was derived by computing the difference between the DSM and DTM, producing a model uninfluenced by the terrain slope and enabling object height determination above ground level [33]. This methodology ensured precise and uniform data representation, supporting subsequent analysis and interpretation.

The reflectance values from each MSP band were used for the spectral VI computation. The selection of VIs was based on their documented effectiveness in the literature

(Appendix A Table A1). Moreover, the thermal VIs were also calculated via TIR imagery, specifically the CWSI and $I_g$ [49], as defined in (1) and (2):

$$CWSI = \frac{T_c - T_{wet}}{T_{dry} - T_{wet}} \tag{1}$$

$$I_g = \frac{T_{dry} - T_c}{T_{dry} - T_{wet}} \tag{2}$$

where $T_c$ represents the plant canopy temperature, and $T_{wet}$ and $T_{dry}$ represent the lower and upper baseline temperatures, respectively, in similar meteorological conditions. The upper baseline characterises the temperature of a non-transpiring leaf with fully sealed stomata, while the lower baseline relates to the leaf temperature with fully opened stomata (undisturbed transpiring leaf). The CWSI was calculated using five distinct methodologies and the $I_g$ by three approaches, as outlined in [22]. For the first four CWSI methodologies ($CWSI_1$, $CWSI_2$, $CWSI_3$ and $CWSI_4$), the $T_{dry}$ and $T_{wet}$ were determined as per Table 2. Notably, the application of the non-water stress baseline (NWSB) insight was recommended for the CWSI computation according to previous studies [50,51]. Therefore, $CWSI_5$ was calculated according to Equation (3), wherein $\Delta T_2$ represents the difference between the canopy and air temperature, while the values of $\Delta T_1$ and $\Delta T_3$ were derived as detailed in Table 2.

$$CWSI_5 = \frac{\Delta T_1 - \Delta T_2}{\Delta T_1 - T_3} \tag{3}$$

**Table 2.** Lower and upper temperature limits for calculating the Crop Water Stress Index (CWSI) and Stomatal Conductance Index ($I_g$).

| Index | Lower Temperature Limit $T_{wet}/\Delta T_1$ | Upper Temperature Limit $T_{dry}/\Delta T_3$ |
|---|---|---|
| $CWSI_1$ | Wet reference temperature | Dry reference temperature |
| $CWSI_2$ | Wet reference temperature | Air temperature plus 3 °C |
| $CWSI_3$ | Wet reference temperature | Severe-water stress canopy temperature |
| $CWSI_4$ | Well-watered canopy temperature | Air temperature plus 3 °C |
| $CWSI_5$ | Temperature difference of the well-watered canopy and air | 3 °C |
| $I_{g1}$ | Wet reference temperature | Dry reference temperature |
| $I_{g2}$ | Wet reference temperature | Air temperature plus 3 °C |
| $I_{g3}$ | Temperature difference of the well-watered canopy and air | 3 °C |

The $I_g$ calculation methods were applied similarly to the prior approach. The $T_{dry}$ and $T_{wet}$ for $I_{g1}$ and $I_{g2}$ were determined following the procedures in Table 2. The $I_{g3}$ approach was computed using Equation (4), with the $\Delta T_1$, $\Delta T_2$, and $\Delta T_3$ being identical to those provided for $CWSI_5$.

$$I_{g3} = \frac{\Delta T_3 - \Delta T_2}{\Delta T_3 - T_1} \tag{4}$$

Two methods were used to estimate the lower and upper reference temperatures. Initially, the temperature of the both upper and lower sides of the water-sprayed leaves ($T_{wet}$) was recorded. Then, the non-transpiring leaves' temperature ($T_{dry}$) was obtained by measuring the leaves covered with petroleum jelly [24]. These measurements established the reference temperatures for further calculations.

Moreover, the average $T_c$ value derived from the TIR imagery of the non-stressed olive trees ($FI_{100}$) served as a reference for the well-watered canopy conditions. This reference temperature was used for the parameter estimations in $CWSI_4$, $CWSI_5$, and $Ig_3$. Conversely, the average $T_c$ of the olive trees near the study area, under rainfed conditions and severe water stress, was used as a dry reference for estimating $CWSI_3$. The air temperature was

monitored using a humidity and temperature data logger (SSN-22, Hairuis Instruments, Shenzhen, China) to ensure precise temperature readings throughout the measurements.

### 2.7. Imagery Segmentation

In remote sensing, the Normalised Difference Vegetation Index (NDVI) is widely applied for segmentation, distinguishing vegetation from non-vegetation areas. In this study, the NDVI was applied to segment and identify olive trees. However, the presence of spontaneous vegetation growing within irrigated rows led to image disturbances (Figure 3c). To address this, the CHM was integrated into the segmentation, considering only heights exceeding 1 m (Figure 3b). This inclusion effectively removed the undesirable vegetation, enhancing the final image quality.

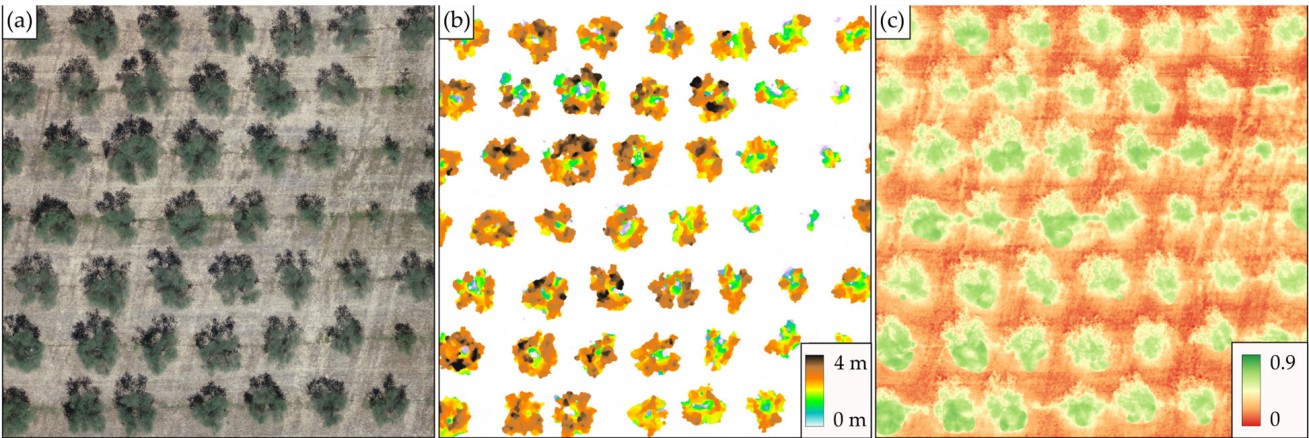

**Figure 3.** Olive orchard example of: (**a**) orthophoto mosaic; (**b**) canopy height model; and (**c**) Normalised Difference Vegetation Index.

Therefore, the process used to generate the binary image for delineating the olive tree crowns included the following stages, as delineated by Marques et al. [19]: (1) implementing Otsu's method [52] on the NDVI image to create a binary representation of the vegetation pixels; (2) integrating CHM into the segment, considering only heights greater than 1 m; (3) merging both binary images; and (4) performing morphological operations such as opening and closing. The resultant image enabled olive tree crown border detection and delineation, allowing for the extraction of the mean values of the spectral and thermal VIs.

### 2.8. Design of the Statistical Analysis and Regression Model Assessment

The data were examined and processed using the statistical software SPSS Statistics 26 (IBM, Armonk, NY, USA). The mean differences in the $\Psi_{MD}$, $g_s$, RWC, and pigment content (Chl *a*, Chl *b*, Chl *ab* and carotenoids) among the irrigation treatments were assessed through conventional analysis of variance (ANOVA) followed by the Tukey HSD test at $p < 0.05$ for significance. Exponential and linear regression equations were established from the VIs against the water status indicators and pigment content. These regression models were then used for the prediction, and the linear correlation between the observed and predicted values was evaluated. The suitability of the regression models and their relation to the original data were assessed using $R^2$. To assess the precision of the linear regression model in predicting the water status indicators and leaf pigment content, three statistical measures—root mean square error (RMSE), mean absolute error (MAE) and relative error (RE)—were used to quantify the deviations between the predicted and observed data.

### 3. Results

#### 3.1. Soil Water Content

Figure 4 illustrates the variation in the $\theta_v$ under the different irrigation strategies during 2019 and 2020. As expected, the reduced water application corresponded to decreased

$\theta$v levels. Notably, among the irrigation regimes, $FI_{120}$, $FI_{100}$, and $RDI_{100}$ (excluding the pit-hardening phase) showed the highest $\theta_v$ values, maintaining relative stability within the 0.25 to 0.32 $cm^3/cm^3$ range. Conversely, $SDI_{30}$ displayed the lowest values (0.13 $cm^3/cm^3$) throughout both years' irrigation seasons. $SDI_{60}$ and $RDI_{60}$ (excluding the pit-hardening phase) displayed intermediate $\theta_v$ values, ranging from 0.15 to 0.20 $cm^3/cm^3$.

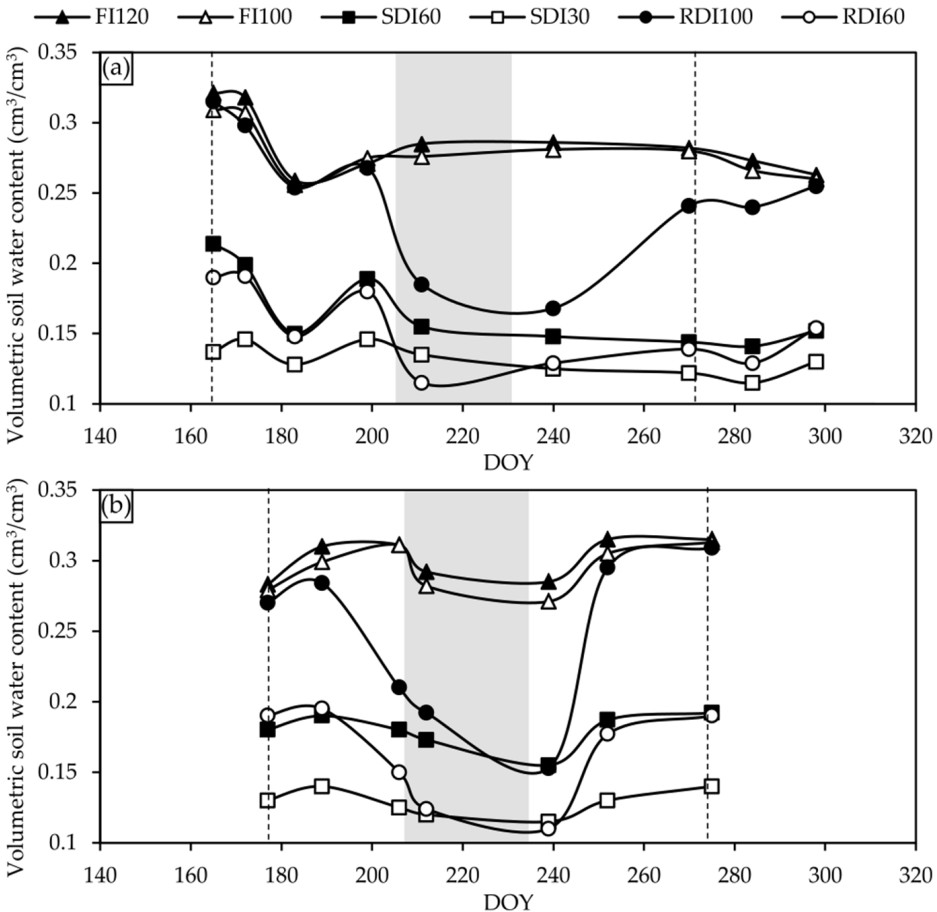

**Figure 4.** Evolution of the mean values of the volumetric soil water content during the irrigation season of 2019 (**a**) and 2020 (**b**) under different irrigation treatments: full irrigated 100% ($FI_{100}$), over full irrigated 120% ($FI_{120}$), sustained deficit irrigation 60% ($SDI_{60}$), sustained deficit irrigation 30% ($SDI_{30}$), regulated deficit irrigation 100% ($RDI_{100}$) and regulated deficit irrigation 60% ($RDI_{60}$). The grey region represents the cutoff irrigation phase in the regulated deficit irrigations and the dashed lines represent the beginning and end of the irrigation season.

Reducing the irrigation volume by 10% in $RDI_{100}$ and completely interrupting the irrigation in $RDI_{60}$ during the pit-hardening phase (from DOY 205 to DOY 232 in 2019 and from DOY 209 to DOY 237 in 2020) rapidly lowered the $\theta_v$ values, aligning with expectations and leading to swift drops in the values. In $RDI_{60}$, the $\theta_v$ slightly dipped below $SDI_{30}$ and approached the soil's lower content capacity, whereas in $RDI_{100}$, the values were slightly above those observed in $SDI_{60}$ during the same period. As for the $\theta_v$ values before the irrigation reduction or interruption, $RDI_{100}$ and $RDI_{60}$ experienced reductions of 60% and 57% in 2019 and 85% and 77% in 2020, respectively. Upon irrigation reestablishment, these treatments recovered, reaching $\theta_v$ levels comparable to similar treatments.

Comparing the minimum $\theta_v$ values observed during the irrigation interruption, $RDI_{100}$ showed a 42% recovery in 2019, while $RDI_{60}$ presented a 21% recovery. In 2020, due to the more significant declines during the cutoff phase, the recovery was more pronounced. Specifically, $RDI_{100}$ achieved a 92% recovery, whereas $RDI_{60}$ recorded a 61% increase in the $\theta_v$.

### 3.2. Water Stress Indicators

Throughout the irrigation season across the three years of the study, measurements of water stress indicators such as the RWC, $\Psi_{MD}$, and $g_s$ were carried out, and the corresponding data are presented in Figure 5. The RWC measurements were conducted before the beginning of the irrigation season on DOY 134 (2019), DOY 145 (2020) and DOY 148 (2021). At this point, no statistically significant differences among the irrigation treatments were detected. The recorded values exceeded 90% of the RWC, indicating non-stressed conditions for the olive trees, consistent with prior research [50]. The RWC values for the $FI_{100}$ and $FI_{120}$ irrigation treatments remained relatively stable, close to 90%, over the three-year irrigation season. Conversely, during the irrigation period, the RWC values gradually declined across all the DI treatments, with significant differences emerging between them. By the end of July in both 2019 and 2020 (DOY 211), $RDI_{60}$, $SDI_{60}$, and $SDI_{30}$ showed notable drops in the RWC. In 2019, these treatments reached minimum RWC values of 82%, 84%, and 77% respectively; in 2020, the corresponding minimum values were 79%, 85%, and 73%, indicating moderate water stress conditions [50]. In 2021, the distinction between the non-stressed and DI treatments emerged earlier, around mid-June (DOY 162), although the values remained stable until late July. Unlike 2019 and 2020, the year 2021 showed minor variations, with the DI recording higher RWC values and lower differences between them.

During the irrigation interruption phase, the RWC values of $RDI_{100}$ and $RDI_{60}$ experienced a significant decline across all the study years. $RDI_{100}$ showed lower values compared to $SDI_{60}$, while $RDI_{60}$ demonstrated lower values than $SDI_{30}$. Within this specific period, $RDI_{100}$ and $RDI_{60}$ indicated minimum values of 83% and 69% in 2019, followed by 83% and 73% in 2020 and 86% and 82% in 2021. A comparison of the RWC values for these irrigation strategies before the interruption period revealed the most significant decrease in 2019, when $RDI_{100}$ and $RDI_{60}$ presented reductions of 9% and 21%, respectively. During this specific period, in 2019 and 2020, $RDI_{60}$ indicated values representing severe water stress [50]. Following the reestablishment of irrigation, a recovery was observed, with $RDI_{100}$ reaching values similar to $FI_{100}$ and $RDI_{60}$ matching the values of $SDI_{60}$. Moreover, in 2019 and 2020, $SDI_{60}$ and $SDI_{30}$ displayed a gradual decrease throughout the irrigation season, except for a minor recovery at DOY 246 in 2020 due to a 39 mm rainfall event recorded during the previous week. By early autumn of 2019 and 2020, $SDI_{30}$ reached minimal RWC values of 64% and 68%, respectively, representing reductions of 28% and 23% compared to the control treatment ($FI_{100}$) on the same DOY.

As carried out in the estimation of the RWC values, the $\Psi_{MD}$ measurements were performed at the beginning of the study, prior to the irrigation starting, except in 2021. At this point, no significant disparities were observed between the irrigation treatments, with the $\Psi_{MD}$ values ranging from −1.7 to −2.1 MPa. However, over the three study years, around DOY 190, $RDI_{60}$, $SDI_{60}$ and $SDI_{30}$ began to manifest disparities with the values of the control treatment. These values showed a mean decrease of 31%, 28% and 38% compared to $FI_{100}$, indicating significant differences.

Notably, in 2020, $\Psi_{MD}$ measurements were not performed during the irrigation interruption phase due to logistical challenges arising from the COVID-19-related limitations on acquiring $N_2$ gas. During this phase, in 2019, a decrease of 50% and 55% was verified in $RDI_{100}$ and $RDI_{60}$, respectively. Similarly, in 2021, the decline reached 54% and 61% in these irrigation strategies. Specifically, the average $\Psi_{MD}$ value for $RDI_{60}$ was −6.2 MPa, representing a 60% decrease compared to the control treatment. Following irrigation reestablishment, the $\Psi_{MD}$ values in $RDI_{100}$ regained comparability to those observed in $FI_{100}$ across all the study years. Conversely, all the DI strategies showed statistical differences from the $FI_{100}$, which were more pronounced in 2019.

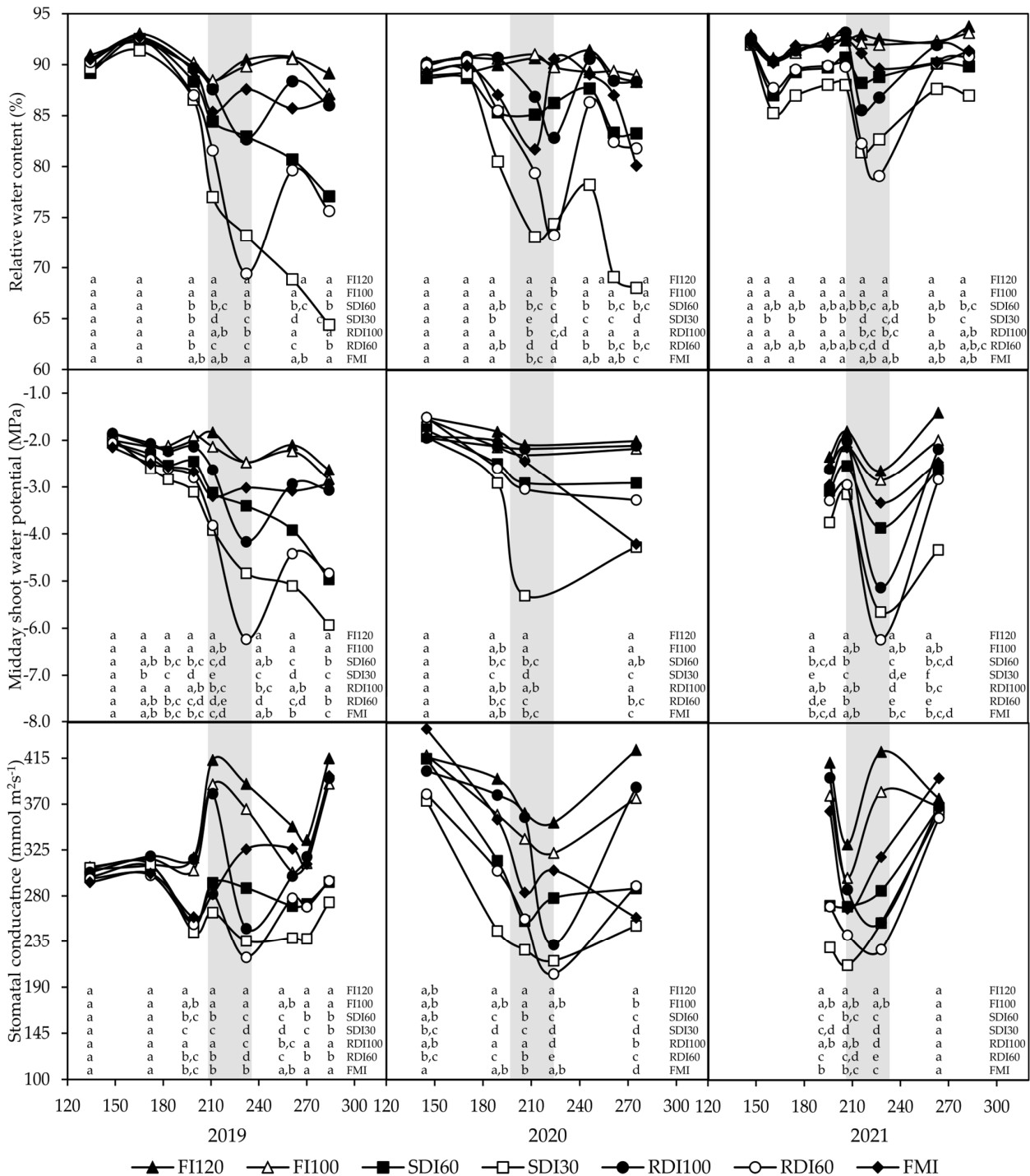

**Figure 5.** Seasonal time course of the relative water content—RWC (*n* = 5), stomatal conductance—$g_s$ (*n* = 3) and midday shoot water potential—$\Psi_{MD}$ (*n* = 3) during the irrigation season from 2019 to 2021 for the different irrigation treatments: full irrigated 100% (FI$_{100}$), over full irrigated 120% (FI$_{120}$), sustained deficit irrigation 60% (SDI$_{60}$), sustained deficit irrigation 30% (SDI$_{30}$), regulated deficit irrigation 100% (RDI$_{100}$), regulated deficit irrigation 60% (RDI$_{60}$) and farmer managed irrigation (FMI). The grey region represents the cutoff irrigation phase in the regulated deficit irrigations. The different lower-case letters represent significant differences between the irrigation treatments within each date (*p* < 0.05).

Among the assessed water stress indicators, the $g_s$ emerged as the parameter revealing the most significant disparities at an earlier stage. At the beginning of July (DOY 185), the

mean $g_s$ value for $SDI_{30}$ was 240 mmol m$^{-2}$s$^{-1}$, representing a decline of 23%, 36%, and 38% in 2019, 2020, and 2021, respectively, in comparison to the control treatment (Figure 5). This outcome suggests that the $g_s$ proved to be the most sensitive indicator of water stress, detecting changes even before the critical periods of elevated temperatures observed in the following months. However, it was also identified as the least consistent indicator, with the values in the non-stressed treatments showing notable fluctuations throughout the irrigation season. During the irrigation interruption phase, the $g_s$ values for $RDI_{100}$ and $RDI_{60}$ experienced noticeable declines, reflecting the patterns observed with the RWC and $\Psi_{MD}$. The reduction was more pronounced in 2019, with $RDI_{100}$ and $RDI_{60}$ showing decreases of 37% and 22%, respectively. Similar to the RWC findings, $RDI_{60}$ demonstrated the lowest $g_s$ value at the end of the irrigation interruption period, with an average of 210 mmol m$^{-2}$s$^{-1}$. This indicated a 35% decrease compared to the control treatment. These observations underscore the potential of the $g_s$ to function as an early indicator of water stress in plants.

### 3.3. Leaf Pigment Content

The assessment of the pigment content revealed notable distinctions among the irrigation treatments in the years 2019 and 2021, as presented in Table 3. Among the analysed pigments, the Chl *a* content showed the most significant disparities, leading to the identification of five distinct homogeneous subsets in 2019. In contrast, the Chl *b* content demonstrated minimal statistical variations between the irrigation strategies, resulting in only two homogeneous subsets in both years. Specifically, in 2019, contrasts were observed exclusively in the $FI_{120}$ treatment when compared to the other treatments.

**Table 3.** Mean values of leaf pigment content by irrigation strategy. Different lower-case letters represent significant differences between the irrigation treatments within each date ($p < 0.05$).

| DOY | Irrigation Strategy | Chl *a* (µg/g) | Chl *b* (µg/g) | Chl *ab* (µg/g) | Carotenoids (µg/g) |
|---|---|---|---|---|---|
| 199 (2019) | $FI_{120}$ | 1244 ± 25 [a] | 589 ± 27 [a] | 1833 ± 14 [a] | 234 ± 7 [a] |
| | $FI_{100}$ | 976 ± 26 [b] | 315 ± 46 [b] | 1290 ± 65 [b] | 128 ± 24 [c] |
| | $SDI_{60}$ | 860 ± 11 [c] | 314 ± 49 [b] | 1173 ± 50 [b] | 192 ± 8 [a,b] |
| | $SDI_{30}$ | 567 ± 14 [e] | 276 ± 12 [b] | 843 ± 24 [c] | 111 ± 12 [c] |
| | $RDI_{100}$ | 690 ± 20 [d] | 241 ± 46 [b] | 932 ± 74 [c] | 149 ± 25 [b,c] |
| | $RDI_{60}$ | 799 ± 22 [c] | 379 ± 49 [b] | 1178 ± 47 [b] | 164 ± 15 [b,c] |
| | FMI | 977 ± 38 [b] | 267 ± 25 [b] | 1244 ± 66 [b] | 206 ± 40 [a,b] |
| 207 (2021) | $FI_{120}$ | 1275 ± 41 [a] | 452 ± 24 [a] | 1727 ± 38 [a] | 353 ± 8 [a,b] |
| | $FI_{100}$ | 1177 ± 42 [a,b] | 469 ± 42 [a] | 1647 ± 53 [a,b] | 361 ± 6 [a,b] |
| | $SDI_{60}$ | 899 ± 17 [d] | 381 ± 8 [a,b] | 1279 ± 20 [c,d] | 330 ± 7 [b] |
| | $SDI_{30}$ | 844 ± 28 [d] | 326 ± 10 [b] | 1179 ± 37 [d] | 288 ± 3 [c] |
| | $RDI_{100}$ | 1073 ± 46 [b,c] | 465 ± 42 [a] | 1537 ± 58 [a,b] | 377 ± 21 [a] |
| | $RDI_{60}$ | 1043 ± 54 [c] | 419 ± 25 [a,b] | 1462 ± 63 [b,c] | 378 ± 17 [a] |
| | FMI | 1085 ± 34 [b,c] | 430 ± 51 [a,b] | 1515 ± 63 [b] | 384 ± 21 [a] |

A comparison of the results between 2019 and 2021 exposed an overall escalation in the pigment content in 2021, with average increases of 21%, 24%, 22% and 108% for the Chl *a*, Chl *b*, Chl *ab* and carotenoids, respectively. These findings aligned with the water stress indicators, particularly the RWC, which indicated that the DI treatments revealed elevated values compared to preceding years, displaying reduced differences among themselves. However, a consistent pattern emerged in both years: a more severe water deficit corresponded to a lower pigment content.

### 3.4. Modelling the Water Stress Indicators and Pigment Content

The VIs listed in Appendix A Table A1 were applied to analyse and predict the water stress indicators, including the RWC, $\Psi_{MD}$ and $g_s$, as well as the pigment content (Chl

*a*, Chl *b*, Chl *ab*, and carotenoids). Moreover, linear and exponential regressions were performed to explore the relationships between the previous parameters and several MSP- and TIR-based VIs. The regression model with the highest coefficient of determination is presented in Appendix B Tables A2 and A3. Examples of the mean VI values for the olive trees, categorised by the irrigation treatment, were derived from UAV data acquired 51 days after the beginning of the irrigation season (DOY 199 in 2019), as illustrated in Figure 6.

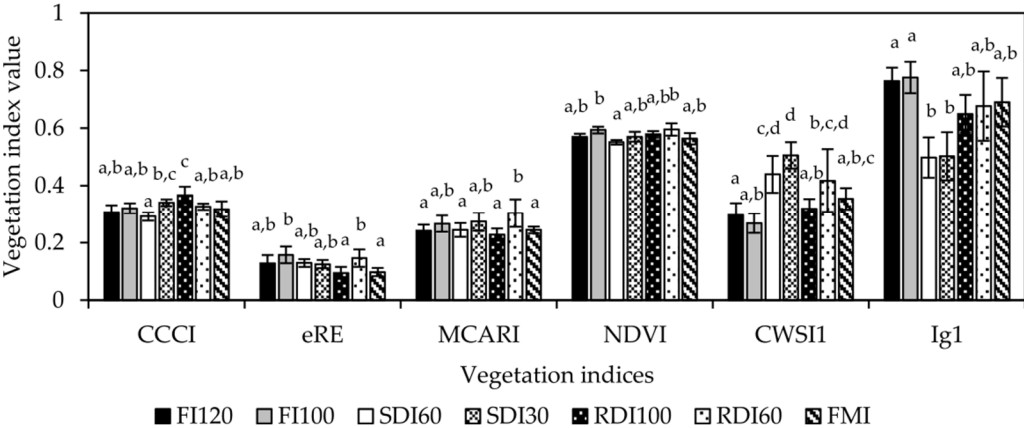

**Figure 6.** Mean values of the various vegetation indices (*n* = 5) on DOY 199 (2019) by the irrigation treatment: full irrigated 100% (FI$_{100}$), over full irrigated 120% (FI$_{120}$), sustained deficit irrigation 60% (SDI$_{60}$), sustained deficit irrigation 30% (SDI$_{30}$), regulated deficit irrigation 100% (RDI$_{100}$), regulated deficit irrigation 60% (RDI$_{60}$) and farmer managed irrigation (FMI). The different lower-case letters represent significant differences between the irrigation treatments within each date (*p* < 0.05). The vertical lines represent the standard deviation.

The thermal indices demonstrated greater efficacy in estimating the water stress indicators. This can be attributed to the positive correlation between the $T_c$ and plant water stress and the negative correlation with the value of the water stress indicator. This relationship is illustrated in Figure 7, where higher water stress leads to higher $T_c$ values and lower $g_s$ values. Moreover, despite the noted correlations between the thermal VIs and RWC, with the highest coefficient of determination ($R^2 > 0.8$), the $g_s$ emerged as the indicator with the most Vis, showing a relatively proficient capability for its estimation (Appendix B Tables A2 and A3).

Conversely, the MSP indices demonstrated more robust correlations with the pigment content than the thermal indices. Particularly, the structural and chlorophyll-related vegetation indices (MCARI and Triangular Vegetation Index—TVI) showed higher correlations with the pigment content, being outstanding for the carotenoid and dry matter-related VIs. Interestingly, the carotenoid-related vegetation indices (Anthocyanin Content Index—ACI, Anthocyanin Reflectance Index—ARI, and Modified Anthocyanin Content Index—mACI), as well as the dry matter-related vegetation indices (Browning Reflectance Index—BRI and Red Green Index—RGI), revealed the weakest performance in estimating either the water stress indicators or pigment content. Their correlations were considerably low ($R^2 < 0.3$). Among the pigments analysed, a larger range of VIs demonstrated the capability for accurate estimation of the Chl *ab* content, while no index proved proficient at precisely estimating the carotenoid content ($R^2 < 0.38$).

### 3.4.1. Model Development

This section describes the initial modelling phase, wherein linear and exponential regression models were performed to establish the correlations among the water status indicators and pigment content (VIs). The effectiveness of these models was assessed via their respective $R^2$ values. Afterwards, the regression approach yielding the highest $R^2$ value was selected for prediction purposes in the model application phase. A graphical

representation of the most proficient indices for estimating each water status indicator is illustrated in Figure 8. The results demonstrate that, concerning the estimation of the water stress indicators, the thermal indices ($CWSI_1$, $CWSI_2$ and $CWSI_3$) show substantial correlations with the RWC ($R^2 = 0.85$, 0.74 and 0.70, respectively, $p < 0.01$). Among the MSP indices, the chlorophyll-related index MCARI achieved the best performance ($R^2 = 0.58$, $p < 0.05$). For the estimation of the $\Psi_{MD}$, lower coefficients of determination were observed. Notably, the thermal indices $CWSI_1$ and $I_{g2}$ emerged as the most effective predictors ($R^2 = 0.66$ and 0.64, respectively, $p < 0.05$). As for the $g_s$ estimations, all the variations of the CWSI demonstrated robust performance ($R^2 > 0.75$, $p < 0.01$), while the $I_{g2}$ also yielded significant results ($R^2 = 0.57$, $p < 0.05$). Furthermore, several MSP indices, including the Adjusted Transformed Soil-Adjusted Vegetation Index (ATSAVI), Green Soil-Adjusted Vegetation Index (GSAVI), Enhanced Vegetation Index 2 (EVI2), Excess RedEdge (eRE), Green Difference Vegetation Index (GDVI), Gitelson and Merzlyak Index (GM1), Normalised Difference Excess Red Edge (NDExRE), Optimised Soil-Adjusted Vegetation Index (OSAVI) and Soil-Adjusted Vegetation Index (SAVI), demonstrated the capability to effectively estimate the $g_s$ with an $R^2 > 0.65$. Particularly, the eRE emerged as exhibiting the best performance, with an $R^2$ of 0.71. Remarkably, within these VIs, the GM1 remains the only chlorophyll-related index, while the remaining are structural.

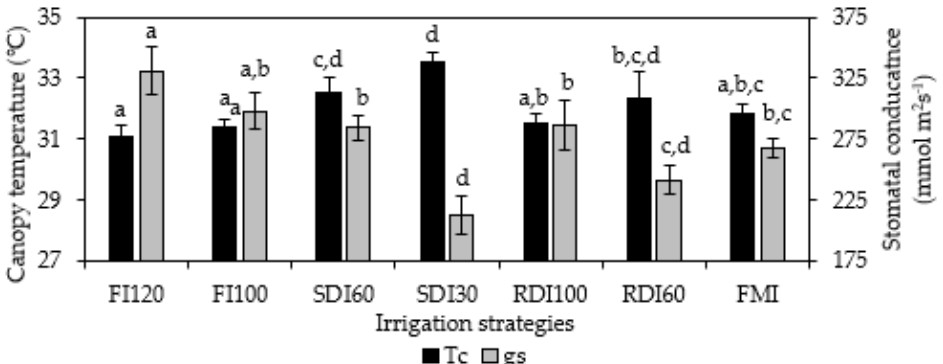

**Figure 7.** Effects of the irrigation treatment on the canopy temperature, $T_c$ (°C), and stomatal conductance, $g_s$ (mmol m$^{-2}$s$^{-1}$), of olive trees. The different lower-case letters represent significant differences between the irrigation treatments ($p < 0.05$): full irrigated 100% ($FI_{100}$), over full irrigated 120% ($FI_{120}$), sustained deficit irrigation 60% ($SDI_{60}$), sustained deficit irrigation 30% ($SDI_{30}$), regulated deficit irrigation 100% ($RDI_{100}$), regulated deficit irrigation 60% ($RDI_{60}$) and farmer managed irrigation (FMI). The vertical lines represent the mean standard deviation.

Regarding the prediction of the pigment content, the chlorophyll-related indices MCARI and TVI showed the best performance, as illustrated in Figure 9. These two VIs revealed closely aligned $R^2$ values, although minor deviations were observed concerning the specific type of pigment. While TVI produced a higher $R^2$ for the Chl *a* ($R^2 = 0.68$, $p < 0.05$) and Chl *ab* ($R^2 = 0.73$, $p < 0.01$), MCARI outperformed it in estimating the Chl *b* ($R^2 = 0.59$, $p < 0.05$). As previously mentioned, no VI achieved good results in estimating the carotenoid content, with TVI yielding the highest coefficient of determination ($R^2 = 0.37$).

### 3.4.2. Model Application

In the second phase of modelling, the regression models developed in the earlier stage were applied to predict the water status indicators and pigment content values. For this purpose, two UAV flights unused in the preceding phase, namely a TIR and an MSP flight on DOY 283 (2018) and an MSP flight on DOY 189 (2020), were used. A detailed overview of the errors associated with the regression models used for predicting the water status indicators and pigment content values is provided in Table 4.

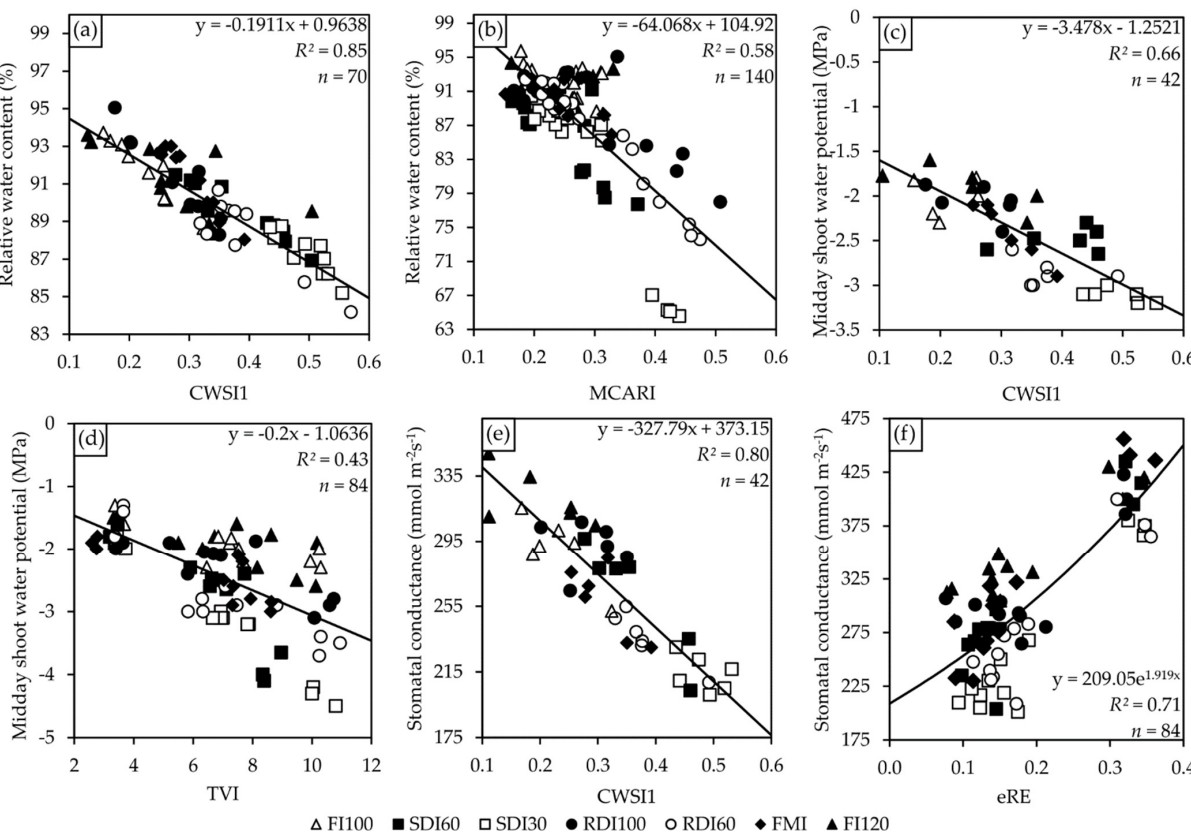

**Figure 8.** Cross-validation between the evaluated indices and water stress indicators: (**a**,**b**) relative water content; (**c**,**d**) midday shoot water potential; and (**e**,**f**) stomatal conductance.

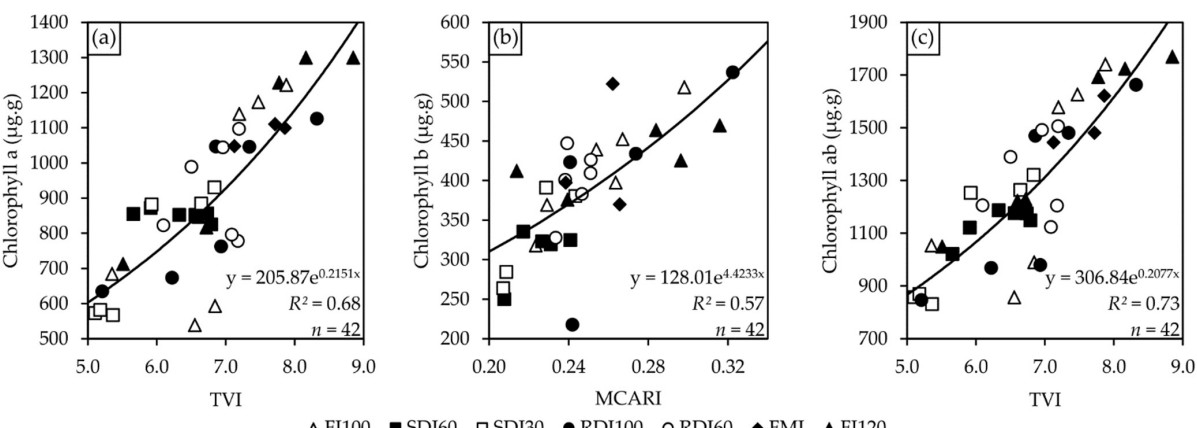

**Figure 9.** Cross-validation between the evaluated indices and pigment content: (**a**) chlorophyll a; (**b**) chlorophyll b; and (**c**) chlorophyll ab.

Consistent with expectations from the model development phase, the thermal VIs demonstrated higher $R^2$ values when predicting water stress indicators such as the RWC, $\Psi_{MD}$ and $g_s$. Conversely, the multispectral VIs showed superior performance in predicting pigment contents such as the Chl *a*, Chl *b* and Chl *ab*. However, even though distinct $R^2$ values were observed between the CWSI$_1$, MCARI and TVI for predicting the RWC and $\Psi_{MD}$, the corresponding MAE values were rather comparable, differing by 0.6% for the RWC and 0.1 MPa for the $\Psi_{MD}$. Nevertheless, more pronounced disparities were evident in the RMSE and RE, emphasising variations among the assessed VIs. Moreover, through the analysis of the RE, it becomes evident that the estimations of the Chl *a* and Chl *ab* showed the lowest values, both below 10%. However, the estimation of the Chl *b* resulted in an RE

of 16%. As expected, due to its limited coefficient of determination value identified during the model selection phase, the prediction of the carotenoids produced substantial errors across all the indicators, showing an RE exceeding 30%. This highlights the lack of accuracy in predicting this particular pigment. Furthermore, Figure 10 graphically represents the correlation between the predicted and observed values of the water status indicators, while Figure 11 illustrates the corresponding correlations for the pigment content.

**Table 4.** Regression characteristics of the observed versus predicted parameters. n: sample number; $R^2$: coefficient of determination; MAE: mean absolute error; RMSE: root mean square error; RE: relative error. The units for the MAE and RMSE vary depending on the predicted parameter: RWC: %; $g_s$: mmol m$^{-2}$s$^{-1}$; $\Psi_{MD}$: MPa; pigment content: μg/g.

| Parameter | Index | Regression Model | n | $R^2$ | MAE | RMSE | RE (%) |
|---|---|---|---|---|---|---|---|
| **Water status indicators** | | | | | | | |
| RWC | CWSI$_1$ | RWC = −0.1911 × CWSI$_1$ + 0.9638 | 35 | 0.80 | 2.3 | 2.7 | 2.8 |
| | MCARI | RWC = −0.6407 × MCARI + 1.0492 | 35 | 0.49 | 2.9 | 3.8 | 3.6 |
| $g_s$ | CWSI$_1$ | $g_s$ = −327.79 × CWSI$_1$ + 373.15 | 35 | 0.72 | 51.7 | 59.7 | 15.1 |
| | eRE | $g_s$ = 209.05 × exp (1.919 × eRE) | 35 | 0.62 | 73.3 | 81.5 | 20.7 |
| $\Psi_{MD}$ | CWSI$_1$ | $\Psi_{MD}$ = −3.478 × CWSI$_1$—1.2521 | 21 | 0.61 | 0.4 | 0.4 | 12.2 |
| | TVI | $\Psi_{MD}$ = −0.2 × TVI—1.0636 | 21 | 0.37 | 0.5 | 0.6 | 20.3 |
| **Pigment content** | | | | | | | |
| Chl *a* | TVI | Chl *a* = 205.87 × exp (0.2151 × TVI) | 21 | 0.61 | 78.3 | 89.2 | 9.8 |
| Chl *b* | MCARI | Chl *b* = 128.01 × exp (4.4233 × MCARI) | 21 | 0.52 | 55.1 | 62.4 | 16.4 |
| Chl *ab* | TVI | Chl *ab* = 306.84 × exp (0.2077 × TVI) | 21 | 0.64 | 103.7 | 116.8 | 9.2 |
| Carotenoids | TVI | Carotenoids = 32.632 × exp (0.2934 × TVI) | 21 | 0.29 | 116.1 | 144.5 | 30.3 |

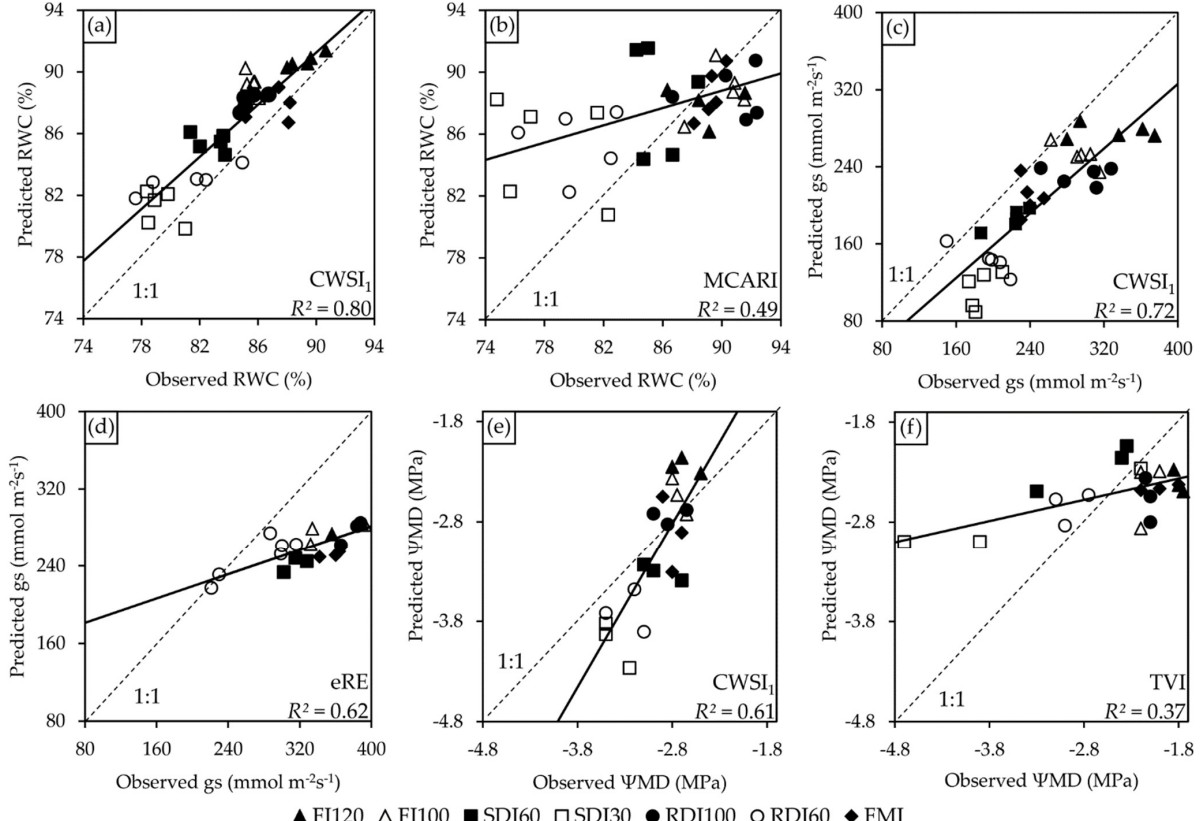

**Figure 10.** Cross-validation scatter plots of the observed water status indicators versus the predicted values derived from the regression models: (**a**,**b**) relative water content; (**c**,**d**) stomatal conductance; and (**e**,**f**) midday shoot water potential.

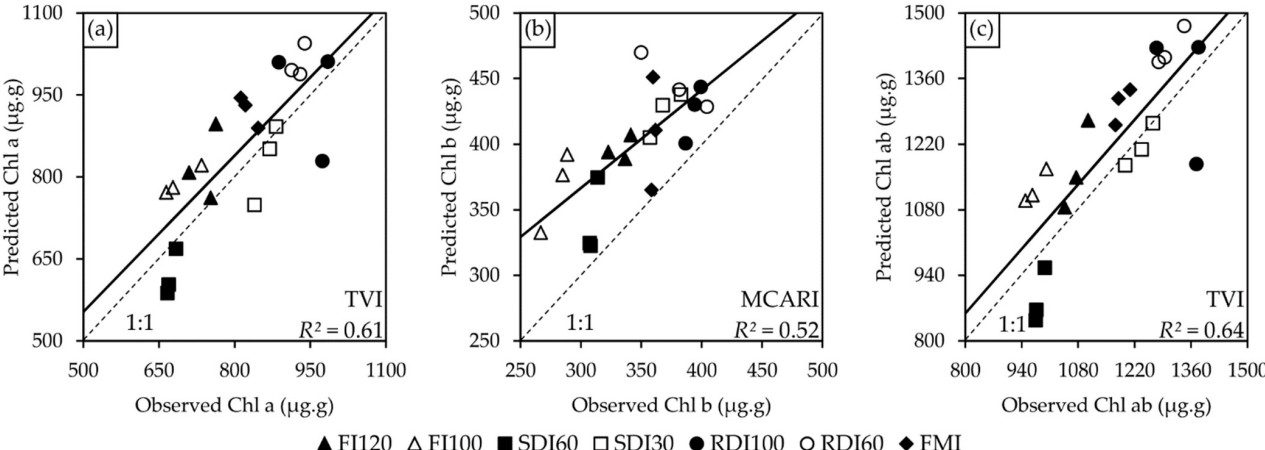

**Figure 11.** Cross-validation scatter plots of the observed pigment content versus the predicted values derived from the regression models: (**a**) chlorophyll a; (**b**) chlorophyll b; and (**c**) chlorophyll ab.

## 4. Discussion

### 4.1. Effects of Water Stress on Pigment Content

A water deficit has a notable influence on the water status of olive trees, as evidenced by the water stress indicators illustrated in Figure 5. Throughout the study period, it became evident that the treatments with a more pronounced water deficit showed considerably lower values for these indicators, with a particularly pronounced decrease observed towards the end of August. The substantial decline in the RWC can be attributed to the increased transpiration rates and inadequate water replenishment. This phenomenon reflects the plant's inability to sustain optimal cellular hydration. Furthermore, the water deficit led to a reduction in the $\Psi_{MD}$ due to the limited availability of soil water. Therefore, the plant restricted water transport from its roots to shoots as a conservation strategy. Additionally, in response to water scarcity, plants enable a regulatory mechanism concerning the $g_s$, leading to stomatal closure. This strategy led to diminished $CO_2$ uptake and reduced water vapor loss. Thus, the rate of photosynthesis decelerated due to the restricted $CO_2$ availability [50,51]. The decrease in the chlorophyll content, an essential pigment for photosynthesis, represents another significant outcome of the water deficit. The reduced water availability restricts chlorophyll synthesis, causing its degradation and the following decline in the pigment content [52,53].

The impact of the water deficit was particularly pronounced in SDI$_{30}$, characterised by sustained elevated water stress throughout the entire irrigation season. This treatment showed consistently low $g_s$ (approximately 200 mmol m$^{-2}$s$^{-1}$) throughout the study's three years, primarily evident at the end of August. This underscores the significant influence of the water deficit on olive leaf stomatal behaviour, detrimentally affecting the chlorophyll content and subsequent photosynthetic processes. In fact, even before reaching minimum conductance levels, the chlorophyll content of the SDI$_{30}$ olive trees was already compromised. Biochemical analysis results from DOY 199 (2019) and DOY 207 (2021) revealed a significantly reduced chlorophyll content compared to the other treatments (Table 3).

As for the leaf pigment content, the chlorophyll and carotenoid content declined under the DI compared to FI strategies. The analysis of the pigment content based on the irrigation strategies presented notable differences between the treatments in 2019 and 2021, as shown in Table 3. Among the analysed pigments, the Chl *a* content showed the most pronounced differences, with five distinct homogeneous subsets in 2019. Conversely, the Chl *b* content showed minimal statistical differences between the irrigation strategies, with only two homogeneous subsets in both years. Specifically, in 2019, only FI$_{120}$ showed contrasts compared to the other treatments. A substantial difference was noted when comparing the most water-limited DI (SDI$_{30}$) with the most well-watered irrigation (FI$_{120}$) in 2019 and 2021.

In 2019, reductions of 54%, 35%, 54% and 53% were observed in the Chl *a*, Chl *b*, Chl *ab* and carotenoid content, respectively. In 2021, the decline was less pronounced, with reductions of 34%, 28%, 32%, and 18% in the corresponding pigments. To clarify the pigment content disparities between the two years, the average air temperature ($T_a$ in °C), vapour pressure deficit (VPD in kPa) and $ET_0$ (mm) were analysed 15 days prior to the leaf collection for biochemical analysis. Upon comparison, it was concluded that, in 2019, the average $T_a$ and $ET_0$ were higher, while the VPD was lower in 2021. Specifically, 2019 showed an average $T_a$ of 25.9 °C, VPD of 2.71 kPa and $ET_0$ of 5.5 mm, while 2021 recorded values of 25.5 °C, 2.30 kPa and 5.4 mm, respectively. Furthermore, 2019 recorded the highest minimum and maximum values for these parameters within the defined timeframe compared to 2021. Thus, these climatic fluctuations between the years may account for the observed pigment content differences.

In general, the consistent declines in the pigment content observed in this study, as reported in previous studies, may arise from the inhibition of chlorophyll biosynthesis or degradation of existing chlorophyll molecules. Additionally, the overall chlorophyll content reduction could be attributed to either degradation or deficiency in synthesis, coupled with changes in the thylakoid membrane structure [54]. However, as highlighted by Marques et al. [18], the spectral response of olive tree leaves varies based on the cultivar, growth stage, and water stress severity. Therefore, comprehending the mechanisms underlying the plant's response to water stress is essential in developing effective strategies to mitigate the adverse impacts of this abiotic stressor on plant growth and productivity.

### 4.2. Effects of Water Stress on Leaf Reflectance and Vegetation Indices

Nowadays, in the precision agricultural management sector, the integration of multispectral and thermal sensors has become essential. These sensors provide critical data for informed decision-making, offering a complete perspective on crop health and environmental conditions [16]. Thus, farmers and agronomists can optimise resource allocation and enhance agricultural yields. MSP sensors offer wavelengths beyond the human eye's capacity, which can provide precise vegetation health assessments through reflected light and VIs, providing insights into factors such as plant stress, nutrient deficiencies and diseases [16]. This knowledge can, for instance, serve as a basis for targeted interventions, significantly reducing the over-application of pesticides and fertilisers. Conversely, thermal sensors outperform MSP sensors in detecting temperature variations throughout fields. These sensors yield indispensable data on soil moisture levels, irrigation efficiency and crop water stress. Crucially, thermal sensors play a key role in agricultural practices by enabling the early detection of potential issues, thereby allowing farmers to mitigate crop losses and efficiently manage their water resources [36]. Moreover, the combination of MSP and thermal sensors demonstrated in this study provides vital data. The integration of vegetation health data from MSP sensors with thermal insights provides farmers with a complete overview of crop conditions. This approach simplifies data-driven decisions concerning irrigation scheduling, pest management and the optimal timing of harvesting [16,36].

Although the sensors used in this study do not enable continuous spectral reflectance data collection from leaves (as performed by spectrometers or hyperspectral imaging sensors), a comparison was conducted between the crown reflectance of the olive trees under the two extreme irrigation strategies ($SDI_{30}$ vs. $FI_{120}$) on DOY 199 (2019) using four wavelengths provided by the Parrot Sequoia sensor. It was evident that the olive trees under $SDI_{30}$ revealed the highest spectral reflectance across all four bands. In the absence of stress, the olive tree canopies showed mean reflectance values of 9.4%, 9.3%, 23.2% and 32.5% in the green, red, red edge, and NIR bands, respectively. Conversely, under stressed conditions ($SDI_{30}$), the corresponding average values were 9.5%, 10.3%, 23.9% and 33.8%, reflecting an increase of 1%, 11%, 3%, and 4% in these respective bands in comparison to the non-stressed plants. Specifically, the red spectrum displayed notably higher reflectance levels under significant water stress. These results align with the findings reported in [18] for the

same cultivar, although there are some discrepancies. In their study, the authors used a spectroradiometer to investigate the effects of DI on leaf reflectance. As verified in our study, an increase in leaf reflectance was observed in the DI treatments. However, in contrast to our findings, where the red and NIR bands showed more significant percentual differences when compared to the reflectance of the olive trees under FI, in [18] the results presented the greatest disparities occurring in the green and NIR bands. Using different crops, investigations have demonstrated that water stress induces variations in spectral reflectance, depending on the plant and specific wavelengths. For example, in crops such as wheat [55], potato [56], barley [57], sunflower [58] and corn leaves [59], water or nutrient stress tends to increase reflectance in the green and red bands. This increase is attributed to the decreased chlorophyll concentration, leading to reduced radiation absorption. Conversely, in other studies including several plants, such as Bermuda grass [25], *Cinnamomum camphora* [60], citrus [20], maize [61], cotton [62] and pine leaves [63], a water deficit has caused an overall increase in the leaf spectral reflectance across the visible and NIR wavelength ranges. In the context of olive trees, this effect was also confirmed, as the water-stressed olive trees showed higher reflectance values [64].

In this study, variations in reflectance due to the different irrigation strategies resulted in the distinct responses of the several analysed VIs, characterised by positive, negative and neutral correlations between the water stress and VI values. While a subset of VIs demonstrated higher values under the FI strategies compared to the DI, other VIs revealed an opposite effect, with higher values under the DI strategies than the FI treatments. Conversely, a few VIs showed minimal differences between these two strategies, indicating similar values. These distinct behaviours observed among the VIs are dependent upon the specific bands used in their formulas and their interrelations. In other investigations concerning the use of VIs in olive trees, Sun et al. [65] concluded that the Relative Depth Index (RDI), Water-Correlated Reflectance Index (WCRI), Water Index (WI) and Photochemical Reflectance Index (PRI) tend to decrease with lower $g_s$ and RWC values. This aligns with the findings of Boshkovski et al. [66], who analysed the impacts of a water deficit on three vegetation indices (NDVI, WI, and PRI) in three Greek olive cultivars. Among the studied VIs, the WI manifested higher sensitivity to water stress, while the PRI demonstrated comparatively minor disparities. Nevertheless, with the exception of the NDVI, it was not feasible to use the aforementioned VIs in this study due to the incompatibility between the spectral bands used in their formulation and the bands provided by the Parrot Sequoia sensor employed in our research.

Regarding the impacts of water stress on the thermal VIs, it was observed that the olive trees exposed to reduced water availability showed higher $T_c$ values. This response can be predominantly attributed to the closure of the stomata induced by water stress, as discussed in the preceding section. Stomatal activity plays an essential role in plant cooling through transpiration. However, when stomatal activity reduces due to stomatal closure, the loss of water via transpiration decreases, consequently leading to a rise in the leaf temperature [50]. As a consequence, this leads to higher values of the CWSI and lower values of the $I_g$. These results are related to the divergent implications of the values of these VIs, which are outcomes of their unique calculation variables. These outcomes align with those reported by Marques et al. [33]. In their study, the authors conducted three types of temperature analyses using TIR imagery, focusing on the pixel values of the olive canopy, irrigation line and treatment block. Their results consistently demonstrated that the DI treatments revealed higher temperatures than the FI treatments across all three approaches, in which the most substantial temperature difference was observed in the canopy analysis, as performed in our study. However, the authors exclusively investigated the impact of DI on the olive tree canopy temperature, without investigating either thermal or optical VIs. Furthermore, no regression analyses were performed to assess the relationships between the canopy temperature and water status indicators.

### 4.3. Model Performance in Estimating the Water Status Indicators

The relationships between the thermal and optical VIs and water stress indicators (RWC, $g_s$ and $\Psi_{MD}$) were explored using multiple linear and exponential correlations. While thermal VIs have traditionally been used for estimating water stress in olive trees, this study also explored the correlations of optical VIs with the aforementioned indicators. The findings demonstrated promising outcomes, with the CWSI showing superior performance in predicting the values of all the analysed water stress indicators (as illustrated in Table 4). Among the five variations of the CWSI evaluated, $CWSI_1$ presented the highest concordance with the ground measurements, yielding an $R^2$ value of 0.80 and an RMSE of 2.7% for predicting the RWC. However, for the $\Psi_{MD}$ (which showed the lowest prediction performance), the $R^2$ value was 0.61 and the RMSE was 0.4 MPa. In contrast, all the different approaches used to calculate the $I_g$ performed inadequately in the model for predicting the desired values (as outlined in Appendix B), leading to their exclusion from the second phase of the study. Among the optical VIs, the MCARI and eRE demonstrated relatively satisfactory correlations with the RWC and $g_s$, respectively. Specifically, the eRE showed an $R^2 = 0.62$ and an RMSE of 81.5 mmol m$^{-2}$s$^{-1}$, which were identical to the values achieved with the CWSI. However, when evaluating the error indicators (MAE, RMSE and RE), the thermal VIs outperformed the optical VIs, displaying lower deviations. The substantial errors observed in the predictions of the optical VIs derived mainly from their inability to discriminate between the intermediate treatments involving moderate water stress, such as $SDI_{60}$ and $RDI_{60}$, and the other irrigation treatments. As illustrated in Figures 8 and 10, the values of these irrigation treatments were generally overestimated, showing similar values to the non-stressed olive trees. Among the water status indicators, the predictions of the RWC values demonstrated the least errors, both in the thermal and optical VIs. This finding was corroborated by the RE values, where the RWC predictions revealed RE values between 2% and 4%, whereas for the other indicators, the RE exceeded 12%. Furthermore, this result offers additional support for the outcomes reported by Marques et al. [47], where a spectroradiometer was used to estimate the VIs, revealing that the RWC estimation achieved the highest accuracy and performance. This higher accuracy might be attributed to the responsiveness of the water indicator to external factors. Remarkably, the $g_s$ and $\Psi_{MD}$ measurements taken at solar noon emerged as highly sensitive indicators of various factors, such as the wind patterns, solar radiation intensity and thermal fluctuations during the measurement sessions. The positive results observed when predicting the $\Psi_{MD}$ by Caruso et al. [36], wherein measurements were taken after blocking the transpiration of leaves located near the main scaffolds of the tree, suggest potential changes for future investigations using this methodology. Additionally, it is important to acknowledge that the load of the olive tree itself could be a contributing factor impacting the behaviour of these two indicators.

### 4.4. Model Performance in Estimating the Pigment Content

The outcomes presented in this study confirm that while the optical VIs were more effective in estimating the pigment content, the thermal VIs more accurately estimated the water stress indicators (as illustrated in Figures 10 and 11). Prior investigations have explored the potential of aerial imagery to estimate the pigment content of olive trees, although using distinct platforms and sensors. In the study by Zarco-Tejada et al. [41], the researchers used a manned aerial vehicle to capture hyperspectral imagery in an olive orchard of Cv. Arbequina and correlated multiple indices only with the Chl *ab*. The authors concluded that the most effective VIs for estimating the Chl *ab* were the TCARI ($R^2 = 0.6$), MCARI ($R^2 = 0.64$), TCARI/OSAVI ($R^2 = 0.48$) and MCARI/OSAVI ($R^2 = 0.69$). The use of hyperspectral sensors by the authors expanded the spectrum of available bands for calculating VIs beyond the scope of our study. For instance, computing the TCARI was not feasible within our research material. Nevertheless, despite this drawback, the use of MSP sensors, as implemented in our study, presents a considerably more cost-effective alternative when compared to hyperspectral sensors. For small-scale farmers considering

the adoption of remote sensing techniques to enhance the management and monitoring of their olive orchards, the use of hyperspectral sensors becomes impractical due to their high cost. Consequently, in alignment with our findings, both thermal and multispectral sensors yield satisfactory results in estimating the water stress indicators and pigment content. This provides a more economically feasible solution for small-scale farmers. Conversely, Berni et al. [67] used a modified UAV with a six-band MSP camera (MCA-6 Tetracam) and determined the TCARI/OSAVI to be effective in predicting the Chl *ab* ($R^2$ = 0.89 and RMSE of 4.2 μg/cm$^2$). However, the authors exclusively conducted correlation analyses between the VIs and Chl *ab*, without directly measuring the content through biochemical analyses, as was performed in the present study.

In this investigation, the correlations between several optical VIs and the pigment content of olive trees were expansively explored, including the Chl *ab*, Chl *a*, Chl *b* and carotenoids. Due to their low coefficients of determination ($R^2$ < 0.3), as shown in Appendix B, the thermal VIs was not used to estimate the pigment content. In the model creation phase, several VIs revealed higher performance in estimating the Chl *ab* compared to the other pigments. Particularly, the ATSAVI, DVI, EVI2, MCARI, MCARI1, MCARI2, MSAVI, MTVI1, SAVI and TVI emerged as promising indicators for estimating this pigment content, with the MCARI and TVI demonstrating the highest correlations ($R^2$ > 0.65), as illustrated in Figure 9. Nevertheless, for estimating the Chl *a* and Chl *b*, only the MCARI and TVI yielded promising results. In the model application phase, the TVI outperformed the MCARI in estimating the pigment content, achieving $R^2$ values > 0.6. Moreover, both the MCARI and TVI displayed a tendency to overestimate the pigment content, particularly in the FI treatments (Figure 11). None of the VIs demonstrated favourable performance in estimating the carotenoid content. This could be attributed to the fact that carotenoids reflect more in the blue wavelength spectrum, which is not supported by the sensor used in this study, whereas chlorophylls predominantly reflect in the green and red wavelengths [14]. Consistent findings reported by Marques et al. [18] support these results, although the aforementioned authors explored a spectrometer, laboratory equipment, and singular leaf samples to analyse the relationship between the pigments in olive leaves and VIs in the same cultivar. Nevertheless, the performance of different VIs in estimating the pigment content can be subject to variability based on diverse factors, including the cultivar, phenological stage, water stress, and fertilisation. These factors, as corroborated by our study and supported by existing studies specific to olive trees [18,68–72], highlight the characteristic variability in VIs' performance in relation to such estimations.

## 5. Conclusions

The aim of this work was to assess the applicability of thermal and optical VIs for estimating the water stress indicators and pigment content of olive trees of the Cv. Cobrançosa using UAV-based aerial imagery. The study included diverse irrigation treatments, involving FI and DI strategies, aiming to investigate their influence on the water stress indicators, pigment concentrations and spectral reflectance of olive tree leaves. The findings revealed that DI imposed an adverse effect on the chlorophyll and carotenoid content, resulting in reduced values. Additionally, DI induced variations in the leaf spectral reflectance, causing higher reflectance across all the spectral wavelengths studied (green, red, red edge and NIR).

In general, the thermal VIs demonstrated greater efficacy in estimating the water stress indicators, whereas the optical VIs showed superior performance in predicting the pigment content. Specifically, the $CWSI_1$ displayed greater accuracy in predicting the RWC, $\Psi_{MD}$ and $g_s$. The MCARI showed superior performance in estimating the Chl *b*, while the TVI outperformed it in terms of the Chl *a* and Chl *ab* estimation. The MCARI and eRE also demonstrated promising outcomes in estimating the RWC and $g_s$, respectively. However, all the VIs revealed limited performance in estimating the carotenoids due to the absence of the blue spectrum. Consequently, this study emphasises the potential and feasibility of both vegetation-related and thermal VIs in estimating distinct parameters, thus proving to be a valuable tool for irrigation management and crop monitoring in olive orchards.

As future work, it is intended to integrate sensors capable of capturing the blue band, thereby enhancing the carotenoid estimation and expanding the range of tested VIs. This increase would allow exploration across a broader spectrum of wavelengths, thereby supplementing the study's scope. Moreover, the use of UAVs equipped with hyperspectral sensors would provide substantial advantages by enabling continuous wavelength capture, enabling the calculation of more precise VIs for specific parameters. However, given the extensive range of VIs involved, using techniques such as machine learning would be required to discriminate the optimal combination of VIs for each estimation. As technology advances further, the importance of these sensors and approaches in precision agriculture continues to grow, highlighting their essential role in shaping the future of sustainable farming practices.

**Author Contributions:** Conceptualisation, P.M. and A.F.-S.; methodology, P.M. and A.F.-S.; software, P.M.; validation, P.M., L.P. and A.F.-S.; formal analysis, P.M. and L.P.; investigation, P.M., L.P., J.J.S. and A.F.-S.; resources, J.J.S. and A.F.-S.; data curation, P.M. and L.P.; writing—original draft preparation, P.M.; writing—review and editing, L.P., J.J.S. and A.F.-S.; visualisation, P.M. and L.P.; supervision, J.J.S. and A.F.-S.; project administration, A.F.-S.; funding acquisition, A.F.-S. All authors have read and agreed to the published version of the manuscript.

**Funding:** This work was funded by the Project Olive Oil Operational Group—SustentOlive: Improvement of irrigation and fertilization practices at olive farms in Trás-os-Montes for its sustainability (PDR2020 101-032178), financed by the European Agricultural Fund for Rural Development (EAFRD) and the Portuguese State under Ação 1.1 "Grupos Operacionais", integrada na Medida 1. "Inovação" do PDR 2020—Programa de Desenvolvimento Rural do Continente. It was also financed by Project SOIL O-LIVE—The Soil Biodiversity and Functionality of Mediterranean Olive Groves: A Holistic Analysis of the Influence of Land Management on Olive Oil Quality and Safety, Funded by the European Commission under Food, Bioeconomy Natural Resources, Agriculture and Environment, Grant agreement ID: 101091255. This research activity was supported by national funds from the FCT—Portuguese Foundation for Science and Technology under the projects UIDB/04033/2020 and LA/P/0126/2020.

**Data Availability Statement:** The data that support the findings of this study are available from the corresponding author upon reasonable request.

**Acknowledgments:** Pedro Marques acknowledges the financial support provided by national funds through the FCT—Portuguese Foundation for Science and Technology (PD/BD/150260/2019) under the Doctoral Programme "Agricultural Production Chains—from fork to farm" (PD/00122/2012) and from the European Social Funds and the Regional Operational Programme Norte 2020. The authors are very grateful to António Ribeiro for the climate data. In memory of Manuel António Afonso: the authors are very grateful to farmer Manuel António Afonso for allowing this study to be developed.

**Conflicts of Interest:** The authors declare no conflict of interest. The funders had no role in the design of the study; in the collection, analyses, or interpretation of data; in the writing of the manuscript; or in the decision to publish the results.

## Appendix A

**Table A1.** Spectral vegetation indices used in this study and their respective equations.

| Acronym | Name | Sensitivity | Equation | Ref. |
|---|---|---|---|---|
| ACI | Anthocyanin Content Index | Carotenoid | $\frac{G}{N}$ | [73] |
| ARI | Anthocyanin Reflectance Index | Carotenoid | $\frac{1}{G} - \frac{1}{R}$ | [74] |
| ATSAVI | Adjusted Transformed Soil-Adjusted Vegetation Index | Structure | $1.22 \times \frac{N - 1.22 \times R - 0.03}{1.22 \times N + R - 1.22 \times 0.03 + 0.08 \times (1 + 1.22^2)}$ | [75] |
| BRI | Browning Reflectance Index | Dry matter/pigment | $\frac{\frac{1}{G} - \frac{1}{R}}{N}$ | [76] |
| CCCI | Canopy Chlorophyll Content Index | Chlorophyll | $\frac{\frac{N - RE}{N + RE}}{\frac{N - R}{N + R}}$ | [77] |
| CIG | Chlorophyll Index Green | Chlorophyll | $\frac{N}{G} - 1$ | [78] |

**Table A1.** *Cont.*

| Acronym | Name | Sensitivity | Equation | Ref. |
|---|---|---|---|---|
| CIRE | Chlorophyll Index RedEdge | Chlorophyll | $\frac{N}{RE} - 1$ | [79] |
| CVI | Chlorophyll Vegetation Index | Chlorophyll | $N \times \frac{R}{G^2}$ | [80] |
| DVI | Difference Vegetation Index | Structure | $N - R$ | [81] |
| EVI2 | Enhanced Vegetation Index 2 | Structure | $2.4 \times \frac{N-R}{N+R+1}$ | [82] |
| eNIR | Excess NIR | Structure | $2 \times N_n - G_n - R_n - RE_n$ | [83] |
| eRE | Excess RedEdge | Structure | $2 \times RE_n - G_n - R_n - N_n$ | [83] |
| GDVI | Green Difference Vegetation Index | Structure | $N - G$ | [84] |
| GM1 | Gitelson and Merzlyak Index | Chlorophyll | $\frac{RE}{G}$ | [85] |
| GNDVI | Green Normalised Difference Vegetation Index | Chlorophyll | $\frac{N-G}{N+G}$ | [86] |
| GRNDVI | Green–Red NDVI | Structure | $\frac{N-(G+R)}{N+(G+R)}$ | [87] |
| GRVI | Green–Red Vegetation Index | Structure | $\frac{G-R}{G+R}$ | [81] |
| GSAVI | Green Soil-Adjusted Vegetation Index | Structure | $\frac{N-G}{N+G+0.5} \times 1.5$ | [84] |
| IPVI | Infrared Percentage Vegetation Index | Structure | $\frac{N}{N+R} \times (NDVI + 1)$ | [88] |
| mACI | Modified Anthocyanin Content Index | Carotenoid | $\frac{N}{G}$ | [89] |
| MCARI | Modified Chlorophyll Absorption in Reflectance Index | Chlorophyll | $[(RE - R) - 0.2 \times (RE - G)] \times \frac{RE}{R}$ | [42] |
| MCARI1 | Modified Chlorophyll Absorption in Reflectance Index 1 | Structure | $1.2 \times (2.5 \times (N - R) - 1.3(N - G))$ | [90] |
| MCARI2 | Modified Chlorophyll Absorption in Reflectance Index 2 | Structure | $\frac{1.5 \times [2.5 \times (N-G) - 2.5 \times (R-G)]}{\sqrt{(2 \times N+1)^2 - (6 \times N - 5 \times \sqrt{R}) - 0.5}}$ | [90] |
| mGRVI | Modified Green Red Vegetation Index | Structure | $\frac{G^2 - R^2}{G^2 + R^2}$ | [26] |
| MSAVI | Modified Soil-Adjusted Vegetation Index | Structure | $\frac{2 \times N+1 - \sqrt{(2 \times N+1)^2 - 8 \times (N-R)}}{2}$ | [91] |
| MSR N/R | Modified Simple Ratio NIR/RED | Structure | $\frac{\left(\frac{N}{R}\right) - 1}{\sqrt{\left(\frac{N}{R}\right)} + 1}$ | [92] |
| MTVI1 | Modified Triangular Vegetation Index 1 | Structure | $1.2 \times (1.2 \times (N - G) - 2.5 \times (R - G))$ | [90] |
| NDExNIR | Normalised Difference Excess NIR | Structure | $\frac{2 \times N_n - G_n - R_n - RE_n}{2 \times N_n + G_n + R_n + RE_n}$ | [83] |
| NDExRE | Normalised Difference Excess Red Edge | Structure | $\frac{2 \times RE_n - G_n - R_n - N_n}{2 \times RE_n + G_n + R_n + N_n}$ | [83] |
| NDRE | Normalised Difference NIR/RedEdge | Structure | $\frac{N-RE}{N+RE}$ | [77] |
| NDVI | Normalised Difference Vegetation Index | Structure | $\frac{N-R}{N+R}$ | [40] |
| OSAVI | Optimised Soil-Adjusted Vegetation Index | Structure | $\frac{1.16 \times (N-R)}{N+R+0.16}$ | [43] |
| RGI | Red Green Index | Dry matter/pigment | $\frac{R}{G}$ | [93] |
| SAVI | Soil-Adjusted Vegetation Index | Structure | $\frac{N-R}{N+R+0.5} \times 1.5$ | [94] |
| TNDVI | Transformed NDVI | Structure | $\sqrt{\frac{N-R}{N+R} + 0.5}$ | [81] |
| TVI | Triangular Vegetation Index | Chlorophyll | $0.5 \times (120 \times (RE - G) - 200 \times (R - G))$ | [95] |
| WDRVI | Wide Dynamic Range Vegetation Index | Structure | $\frac{0.1 \times N - R}{0.1 \times N + R}$ | [96] |

Note: G: green; R: red; N: NIR; RE: red edge; L = 0.5; $G_n$: $\frac{G}{(N+G+R)}$; $R_n$: $\frac{R}{(N+G+R)}$; $N_n$: $\frac{N}{(N+G+R)}$; $RE_n$: $\frac{RE}{(N+G+R)}$.

## Appendix B

**Table A2.** Coefficients of determination ($R^2$) of the linear relationships between the water status indicators ($n = 132$), leaf pigments ($n = 66$) and spectral vegetation indices (VIs) at DOY 199 and 261 (2019), DOY 145 (2020) and DOY 207 (2021). The highest significant index for each variable is highlighted in bold.

| VIs | RWC | $g_s$ | $\Psi_{MD}$ | Chl *a* | Chl *b* | Chl *ab* | Carotenoids |
|---|---|---|---|---|---|---|---|
| ACI | 0.15 | 0.36 | 0.38 | 0.09 | 0.04 | 0.08 | 0.02 |
| ARI | 0.14 | 0.01 | 0.08 | 0.09 | 0.09 | 0.11 | 0.07 |
| ATSAVI | 0.17 | 0.66 | 0.30 | 0.40 | 0.44 | 0.48 | 0.32 |
| BRI | 0.07 | 0.30 | 0.14 | 0.08 | 0.08 | 0.10 | 0.07 |
| CCCI | 0.02 | 0.64 | 0.07 | 0.30 | 0.18 | 0.28 | 0.06 |
| CIG | 0.14 | 0.44 | 0.40 | 0.05 | 0.03 | 0.05 | 0.08 |
| CIRE | 0.02 | 0.57 | 0.04 | 0.13 | 0.05 | 0.10 | 0.01 |
| CVI | 0.23 | 0.27 | 0.36 | 0.14 | 0.09 | 0.14 | 0.04 |
| DVI | 0.21 | 0.63 | 0.35 | 0.38 | 0.41 | 0.46 | 0.33 |
| EVI2 | 0.20 | 0.65 | 0.33 | 0.41 | 0.43 | 0.49 | 0.34 |
| eNIR | 0.12 | 0.21 | 0.37 | 0.09 | 0.11 | 0.11 | 0.10 |
| eRE | 0.03 | **0.71** | 0.30 | 0.13 | 0.17 | 0.15 | 0.05 |
| GDVI | 0.18 | 0.66 | 0.33 | 0.32 | 0.36 | 0.39 | 0.31 |
| GM1 | 0.08 | 0.68 | 0.34 | 0.09 | 0.09 | 0.09 | 0.05 |
| GNDVI | 0.15 | 0.38 | 0.39 | 0.14 | 0.04 | 0.11 | 0.05 |
| GRNDVI | 0.08 | 0.47 | 0.37 | 0.12 | 0.15 | 0.14 | 0.10 |
| GRVI | 0.30 | 0.02 | 0.10 | 0.01 | 0.11 | 0.03 | 0.02 |
| GSAVI | 0.14 | 0.69 | 0.29 | 0.31 | 0.34 | 0.37 | 0.29 |
| IPVI | 0.01 | 0.48 | 0.26 | 0.33 | 0.24 | 0.33 | 0.19 |
| mACI | 0.14 | 0.43 | 0.40 | 0.09 | 0.03 | 0.07 | 0.02 |
| MCARI | **0.58** | 0.18 | 0.29 | 0.58 | **0.59** | 0.66 | 0.31 |
| MCARI1 | 0.25 | 0.60 | 0.36 | 0.41 | 0.43 | 0.49 | 0.32 |
| MCARI2 | 0.21 | 0.62 | 0.32 | 0.40 | 0.44 | 0.48 | 0.33 |
| mGRVI | 0.30 | 0.02 | 0.10 | 0.12 | 0.17 | 0.16 | 0.07 |
| MSAVI | 0.20 | 0.64 | 0.33 | 0.40 | 0.44 | 0.49 | 0.33 |
| MSR N/R | 0.01 | 0.49 | 0.26 | 0.33 | 0.32 | 0.36 | 0.21 |
| MTVI1 | 0.25 | 0.60 | 0.36 | 0.41 | 0.43 | 0.49 | 0.32 |
| NDExNIR | 0.13 | 0.08 | 0.33 | 0.09 | 0.10 | 0.10 | 0.10 |
| NDExRE | 0.02 | 0.70 | 0.29 | 0.13 | 0.17 | 0.15 | 0.06 |
| NDRE | 0.03 | 0.59 | 0.04 | 0.16 | 0.06 | 0.13 | 0.02 |
| NDVI | 0.03 | 0.47 | 0.26 | 0.33 | 0.27 | 0.35 | 0.21 |
| OSAVI | 0.18 | 0.68 | 0.29 | 0.43 | 0.42 | 0.49 | 0.32 |
| RGI | 0.29 | 0.04 | 0.10 | 0.30 | 0.18 | 0.29 | 0.14 |
| SAVI | 0.19 | 0.66 | 0.32 | 0.42 | 0.44 | 0.50 | 0.34 |
| TNDVI | 0.02 | 0.47 | 0.26 | 0.27 | 0.16 | 0.26 | 0.15 |
| TVI | 0.42 | 0.46 | **0.43** | **0.68** | 0.55 | **0.73** | **0.37** |
| WDRVI | 0.03 | 0.49 | 0.26 | 0.32 | 0.32 | 0.36 | 0.19 |

**Table A3.** Coefficients of determination ($R^2$) for the linear relationships between the water status indicators ($n = 66$), leaf pigments ($n = 66$) and thermal vegetation indices (VIs) at DOY 199 (2019) and DOY 207 (2021). The highest significant index for each variable is highlighted in bold.

| VIs | RWC | $g_s$ | $\Psi_{MD}$ | Chl *a* | Chl *b* | Chl *ab* | Carotenoids |
|---|---|---|---|---|---|---|---|
| $CWSI_1$ | **0.85** | **0.80** | **0.66** | 0.28 | **0.27** | 0.24 | **0.06** |
| $CWSI_2$ | 0.74 | 0.76 | 0.58 | **0.29** | 0.25 | **0.26** | 0.05 |
| $CWSI_3$ | 0.70 | 0.75 | 0.58 | 0.27 | 0.25 | 0.24 | 0.04 |
| $CWSI_4$ | 0.66 | 0.75 | 0.50 | 0.23 | 0.21 | 0.21 | 0.03 |
| $CWSI_5$ | 0.68 | 0.77 | 0.51 | 0.20 | 0.19 | 0.17 | 0.02 |
| $I_{g1}$ | 0.38 | 0.57 | 0.34 | 0.08 | 0.06 | 0.13 | 0.01 |
| $I_{g2}$ | 0.43 | 0.63 | 0.46 | 0.07 | 0.06 | 0.09 | 0.01 |
| $I_{g3}$ | 0.42 | 0.67 | 0.45 | 0.07 | 0.06 | 0.08 | 0.01 |

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
