# Peer review of "Assessing the Water Status and Leaf Pigment Content of Olive Trees: Evaluating the Potential and Feasibility of Unmanned Aerial Vehicle Multispectral and Thermal Data for Estimation Purposes"

_remotesensing, doi:10.3390/rs15194777_

Round 1

Reviewer 1 Report

Dear authores,

The work conforms to the quality standards of Remote Sensing. The objectives and the methodology carried out are clearly stated.

The work presents results of interest for the advance in the management of the olive grove, based on the results derived from the study. The study presents a widely detailed methodology, so it can be replicated, in this crop, in other varieties, as well as in other crops.

The weakest point of the work is found in the discussion of the results, where some previous references are included, in similar studies, although the results are mostly commented on again. It is recommended to review this section of the work incorporating a more detailed comparison with previous works.

In addition, an attached file is included, which includes comments that must be reviewed by the authors, before advancing to the next publication phase.

The final evaluation is positive, so the article is acceptable after making minor changes.

Kind regards

Author Response

The authors would like to express their gratitude to the reviewer for providing valuable comments and suggestions that have greatly enhanced the quality of this manuscript.

In response to the feedback regarding the discussion of the results, and in line with suggestions from other reviewers, the Discussion section has undergone a thorough revision. The aim was to make it more concise and enhance the comparison with the findings of the referenced studies.

The comments and suggestions provided in the attached file have been properly considered in the revised version of the manuscript. For all these reasons, the authors believe that the new version of the manuscript meets the expectations of the reviewer.

Reviewer 2 Report

This paper utilized multispectral and thermal data from UAV platforms to predict water status indicators with linear and exponential regression models. The paper presents clear and straightforward explanations, making it easily comprehensible.

Some major comments:

1. The paper's length should be reduces, particularly the sections related to data collection and processing.

2. The paper lacks novelty. Please emphasize the contributions made by this study, considering that the image segmentation method and the prediction models (linear and exponential regression) employed are considered conventional or widely used.

Some minor comments:

1. Please check the format of text and figures from line 336 - 346. 

2. Please check all the hyperlinks in the text. For example line 412, 458, 476, 616 show error.

3. Please explain those legends in Figure 6, for example "a F120 a  a  a  a", also they are overlapped with x axis.

4. Please check the format of line 516 (the word "Figure 7.")

Minor editing of English language required.

Author Response

The authors would like to thank the reviewer by its valuable observations and suggestions. Major changes to the manuscript text are highlighted. Please, find the answers to the comments and suggestions in the attached PDF.

Reviewer 3 Report

The article discusses “Assessing water status and leaf pigment content in olive trees: 2 evaluating the potential and feasibility of UAV multispectral 3 and thermal data for estimation”. This study investigates the feasibility of utilizing spectral and thermal vegetation 14 indices (VIs) for effective irrigation management of olive trees in the Northeast region of 15 Portugal. Through the utilization of unmanned aerial vehicles (UAVs) and the acquisition 16 of thermal infrared and multispectral data, the performance of 37 optical vegetation indi-17 ces and two thermal indices were analyzed to estimate water status indicators and evalu-18 ate leaf pigment content Overall, the manuscript is satisfactory and interesting. However, this paper requires major revisions to be suitable for publication.

Ø  The title of the ms suggests this work is done to evaluate the potential use of Multi-spectral and thermal sensor. How is this useful in agricultural management? This should be picked up in the discussion.

Ø  From the abstract alone, it is hard to understand what was done in this study and with what aim. It should be rewritten. Overall, the motivation of this work needs to come out more clearly.

Ø  The introduction is rather confusing to read. What was the issue, what was done and how flying altitude influence accuracy? This is not very clear at present. In the introduction section, the author should mention the location of the research field and coordinates at the end part.

Ø  The authors should highlight the scope and status of Precision Irrigation in some developing countries.

Ø  The author should addressed some points “What is the influence of socioeconomic factors, agroecological factors, institutional factors, information sources, farmer perception, behavioral and technological factors in the adoption of Precision irrigation by farmers. ?

Ø  In general, the text needs extensive editing in terms of language and style. As it is, it is very hard to follow the content of this manuscript and evaluate it. To make a proper statement on its content this needs revision.

Ø  Authors should attach some drone photo (inside the field) which is used for this research purpose.

Ø  Author should mention the which algorithm were used to remove the soil background and calculate the canopy temperature. I cannot find any equation or formula to calculate the canopy temperature or remove the soil background.

Ø  In the future adoption section, the study results should be more compared with and interpreted referring to similar studies.

Ø Authors made a good conclusion and very interesting recommendations. As final general comment, please make sure to define ALL the acronyms form their first appearance in the paper. Also, all the references MUST BE CHECKED and formatted as required by the journal.

Ø In general, the text needs extensive editing in terms of language and style. As it is, it is very hard to follow the content of this manuscript and evaluate it. To make a proper statement on its content this needs revision.

Author Response

(The authors gave the same response as above.)

Round 2

Reviewer 3 Report

Considering the thoroughness of the revisions and the positive impact on the manuscript's quality, I am pleased to recommend this article for publication in Remote Sensing.